# Differentiable Causal Discovery of Linear Non-Gaussian Acyclic Models Under Unmeasured Confounding

**Yoshimitsu Morinishi**                                    *s7024104@st.shiga-u.ac.jp*
*Graduate School of Data Science, Shiga University; Strategy and Consulting, Data and AI, Accenture Co. Ltd.*

**Shohei Shimizu**                          *shohei-shimizu@biwako.shiga-u.ac.jp*
*SANKEN; The University of Osaka; Shiga University*

**Reviewed on OpenReview:** *https://openreview.net/forum?id=4056*

## Abstract

We propose a score-based method that extends the framework of the linear non-Gaussian acyclic model (LiNGAM) to address the problem of causal structure estimation in the presence of unmeasured variables. Building on the method proposed by Bhattacharya et al. (2021), we develop a method called ABIC LiNGAM, which assumes that error terms follow a multivariate generalized normal distribution and employs continuous optimization techniques to recover acyclic directed mixed graphs (ADMGs). We demonstrate that the proposed method can estimate causal structures, including the possibility of identifying their orientations, rather than only Markov equivalence classes, under the assumption that the data are linear and follow a multivariate generalized normal distribution. Additionally, we provide proofs of the identifiability of the parameters in ADMGs when the error terms follow a multivariate generalized normal distribution. The effectiveness of the proposed method is validated through simulations and experiments using real-world data.

## 1 Introduction

### 1.1 Background and Motivation

Uncovering causal relationships from observational data—often referred to as causal discovery—is a critical objective in numerous disciplines, including life sciences, social sciences, and marketing. While randomized controlled trials (RCTs) are the gold standard for identifying causal effects, they are frequently infeasible in practice due to prohibitive costs or ethical constraints. Therefore, it is essential to develop robust methods for inferring causal structures from observational data.

Many existing algorithms assume the absence of unmeasured variables, implying that all relevant factors are fully observed (Spirtes et al., 2000; Chickering, 2002; Shimizu et al., 2006). For example, the linear non-Gaussian acyclic model (LiNGAM) (Shimizu et al., 2006) demonstrates that causal directions can be uniquely identified when errors are non-Gaussian and no hidden confounders

exist. However, this assumption of no unmeasured variables is often unrealistic. In practical applications such as consumer purchasing behavior analysis, latent factors like income or occupation can significantly confound observed relationships.

To address this issue, acyclic directed mixed graphs (ADMGs) (Richardson & Spirtes, 2002) have been introduced. By including both directed and bidirected edges, ADMGs can represent the more intricate dependencies induced by latent variables, capturing confounding effects that traditional directed acyclic graphs (DAGs) cannot. Additionally, ADMGs support nonparametric equality constraints, such as the Verma constraints (Verma & Pearl, 1990), providing a flexible framework for causal discovery in the presence of hidden confounders.

**Key Point (1):** ADMGs allow us to capture both *direct causal effects* and *latent confounding* within a single graph, which is crucial in many real-world scenarios where some variables are unobserved.

### 1.2 Related Work

Methods for estimating ADMG structures can be broadly classified into three categories:

**Constraint-based methods.** These use repeated conditional independence tests to build partial ancestral graphs (PAGs) (Spirtes et al., 2000). Prominent examples include FCI (Spirtes et al., 2000), RFCI (Colombo et al., 2012), and GFCI (Ogarrio et al., 2016), extended to high-dimensional data in approaches like lFCI (Chen et al., 2021). While effective under sufficient sample sizes, these methods can be sensitive to the multiplicity and order of tests.

**Score-based methods.** By optimizing a global objective (e.g., log-likelihood or BIC), score-based approaches can provide a more coherent framework for model selection (Nowzohour et al., 2017; Bernstein et al., 2020; Chen et al., 2021; Claassen & Bucur, 2022; Ng et al., 2024). However, many existing score-based techniques assume Gaussian error terms, thus limiting their ability to recover strict causal orientations when latent variables induce equivalence classes.

**Hybrid methods.** These combine constraint-based and score-based strategies, aiming to balance local statistical decisions with global model fit. GFCI (Ogarrio et al., 2016) exemplifies this line of research.

Recent developments emphasize strict orientation recovery under latent confounding by leveraging *non-Gaussianity* or *nonlinearity* (Shimizu et al., 2006; Tashiro et al., 2014; Wang & Drton, 2024). Notably, the BANG method (Wang & Drton, 2024) exploits higher-order moments in a constraint-based procedure to identify bow-free ADMGs beyond Markov equivalence classes. Further refining these ideas, Wang & Drton (2024) (see also references therein) incorporate non-Gaussian assumptions into a framework that can recover directions more strictly, though specialized moment conditions are still crucial for direction recovery.

In parallel, Bhattacharya et al. (2021) introduced ABIC, a *continuous score-based* method for learning ancestral, arid, or bow-free ADMGs under the linear Gaussian setting. Although ABIC unifies different structural constraints within a differentiable optimization framework, *strict orientation recovery* remains elusive under Gaussianity. Subsequent work, such as Ng et al. (2024), has fo-

cused on scalable continuous relaxations for large ADMGs, but again, the challenge of uniquely determining causal directions persists without non-Gaussian assumptions.

**Key Point (2):** In Gaussian settings, we can typically only recover up to a Markov equivalence class. Non-Gaussian error assumptions are needed to fully determine directions.

### 1.3 Our Goal and Contributions

Motivated by these developments, we propose **ABIC LiNGAM**, a novel approach for strictly identifying bow-free ADMGs under linear *non-Gaussian* error distributions. Specifically, we assume error terms follow a *multivariate generalized normal distribution* (MGGD), thereby generalizing ABIC (Bhattacharya et al., 2021) beyond Gaussianity. Our primary contributions are:

1. **Orientation Recovery Beyond Gaussian Equivalence.** We show that non-Gaussianity (i.e., $\beta \neq 1$ in MGGD) can enable more precise identification of causal directions in bow-free ADMGs, going beyond the limitations of Gaussian-based methods that recover only up to Markov equivalence classes.

2. **Unified Framework.** When the shape parameter $\beta = 1$ (Gaussian case), our method reduces to the original ABIC scheme, ensuring compatibility with purely Gaussian models while offering a broader scope for non-Gaussian scenarios.

3. **Identifiability Proof.** Building on Brito & Pearl (2002) and Wang & Drton (2024), we rigorously prove identifiability of bow-free ADMGs under MGGD assumptions, solidifying the theoretical foundation of our approach.

4. **Empirical Validation and Scalability.** We conduct extensive simulations (up to 50 variables) and apply our method to real-world social survey data, illustrating higher accuracy than constraint-based (e.g., FCI) and non-Gaussian methods (e.g., BANG). Moreover, the fully differentiable formulation scales effectively to larger graphs.

**Key Point (3):** In summary, we extend a *continuous score-based* approach (ABIC) from Gaussian to non-Gaussian error assumptions, potentially enabling more thorough orientation recovery for bow-free ADMGs.

To our knowledge, **ABIC LiNGAM** is the first *continuous score-based* method that achieves full identification of bow-free ADMGs under linear non-Gaussian assumptions. The remainder of this paper is organized as follows: Section 2 reviews the linear structural equation model, bow-free ADMGs, and the MGGD. Section 3 introduces our differentiable constraints and likelihood-based objective. Section 4 presents the ABIC LiNGAM algorithm and theoretical properties. Sections 5–6 report experimental results on synthetic and real datasets. We conclude in Section 7 with a discussion of future directions and broader implications. Unlike a simple relaxation of assumptions, the key is the introduction of non-Gaussianity, which provides crucial higher-order information. Hence, our framework can strictly identify causal directions in linear models with latent confounders, rather than settling for Markov equivalence classes.

## 2 Problem Definition

### 2.1 Representation by Linear SEM

In this section, we review linear SEMs and their graphical representations. We use uppercase letters (e.g., $X$) to denote variables or nodes in the graph and indexed uppercase letters (e.g., $X_i$) to denote specific variables or nodes. We also use the following standard matrix notation: $A_{ij}$ denotes the element in the $i$th row and $j$th column of matrix $A$, $A_{-i,-j}$ denotes the submatrix of $A$ obtained by removing the $i$th row and $j$th column, and $A_{:,i}$ denotes the $i$th column of $A$.

Additionally, for each vertex $i$ belonging to set $V$, let $\{\mathrm{pa}(i) \mid i \in V\}$ and $\{\mathrm{sp}(i) \mid i \in V\}$ be two families of index sets. The vertex set of $G$ is the index set $V$, and $G$ contains the edge $j \to i$ if and only if $j \in Pa(i)$ and the edge $j \leftrightarrow i$ if and only if $j \in Sp(i)$ (or equivalently, $i \in Sp(j)$). Furthermore, $\{\mathrm{sp}(i) \mid i \in V\}$ satisfies the following symmetry condition: for any $j \in V$, $j \in \mathrm{sp}(i)$ holds if and only if $i \in \mathrm{sp}(j)$. These two families of sets $\{\mathrm{pa}(i) \mid i \in V\}$ and $\{\mathrm{sp}(i) \mid i \in V\}$ define the system of structural equations.

#### 2.1.1 Linear SEM

We consider linear SEMs for $d$ variables, parameterized by a weight matrix $\theta \in \mathbb{R}^{d \times d}$. For each variable $X_i \in X$, the structural equation is

$$X_i = \sum_{j \in \mathrm{pa}(i)} \theta_{ij} X_j + \epsilon_i, \quad i \in V \tag{1}$$

Here, the noise terms $\epsilon_i$ are mutually independent. In this case, $\mathrm{sp}(i) = \emptyset$ for all $i$, since no unmeasured variables exist. The graph $G$ and corresponding binary adjacency matrix $D \in \{0,1\}^{d \times d}$ are defined as follows: An edge $X_j \to X_i$ exists in $G$ if and only if $\theta_{ij} \neq 0$, in which case $D_{ij} = 1$. The graph $G$ is acyclic if and only if $\theta$ can be created as an upper triangular matrix by the permutation of vertex labeling.

**Key Point (4):** Even in a linear SEM, once unmeasured variables are discovered, we may need an ADMG (not just a DAG) to handle latent confounding via bidirected edges.

#### 2.1.2 Linear SEM with unmeasured variables

An observed set of variables is causally insufficient if there exist unmeasured variables that are the ancestors of two or more observed variables in the system. In a linear structural equation model (SEM), these unmeasured variables often manifest as dependencies among the error terms Pearl (2009). Consider a $d$-dimensional random vector $X = (X_1, \ldots, X_d)$ represented by real-valued matrices $\delta, \Omega \in \mathbb{R}^{d \times d}$. For each $X_i$, the structural equation takes the following form:

$$X_i = \sum_{j \in \mathrm{pa}(i)} \delta_{ij} X_j + \epsilon_i, \quad i \in V. \tag{2}$$

Here, $\epsilon = (\epsilon_1, \ldots, \epsilon_d)$ is a vector of error terms with zero mean without loss of generality and is not necessarily Gaussian. Allowing non-Gaussian noise terms accommodates a wider class of underlying distributions and may improve identifiability via higher-order moments or nonsymmetric distributional features Shimizu et al. (2006); Wang & Drton (2024).

In the special case where a given variable $X_i$ has no unmeasured variables, its error term $\epsilon_i$ may be independent of all the others. However, if unmeasured variables influence multiple observed variables, their corresponding error terms become dependent on each other. These dependencies are captured by the matrix $\Omega = \mathbb{E}[\epsilon\epsilon^\top]$, which does not need to be diagonal. For the Gaussian noise, the marginalized distribution of $X$ is a zero-mean multivariate normal with a covariance matrix as follows:

$$\Sigma = (I - \delta)^{-1}\Omega(I - \delta)^{-\top}, \tag{3}$$

And the same covariance structure can be considered for non-Gaussian errors, at least at the level of second moments. In a non-Gaussian setting, higher-order moments and distributional asymmetries can be exploited to identify causal directions and latent structures.

The induced graph $G$ is an ADMG that includes both directed ($\rightarrow$) and bidirected ($\leftrightarrow$) edges. The graph $G$ and associated binary adjacency matrices $D \in \{0, 1\}^{d \times d}$ and $B \in \{0, 1\}^{d \times d}$ are defined as follows: a directed edge $X_j \rightarrow X_i$ exists in $G$ if and only if $\delta_{ij} \neq 0$, in which case $D_{ij} = 1$. A bidirected edge $X_j \leftrightarrow X_i$ exists in $G$ if and only if $\Omega_{ij} \neq 0$ (symmetry ensures $\Omega_{ji} \neq 0$), in which case $B_{ij} = B_{ji} = 1$. In the special case where there are no unmeasured variables, the ADMG reduces to a DAG, and the $B$ matrix is a zero matrix.

In summary, this framework does not restrict the noise terms to be Gaussian, allowing a broader class of SEMs that can represent latent variable-induced dependencies through non-Gaussian distributions. By leveraging non-Gaussianity, one can potentially achieve stronger identifiability and more robust causal inferences than would be possible under Gaussian assumptions alone.

**Key Point (5):** Non-Gaussian error terms can often break Markov equivalences that would hold under purely Gaussian assumptions, thus allowing clearer causal direction recovery.

### 2.2 Motivation Example

As discussed in Section 2.1.1, when there are no unmeasured variables, the observed variables can be represented as a DAG. Thus, the problem reduces to estimating the structure of a DAG. However, when unmeasured variables are present, a DAG cannot adequately represent the relationships between variables while accounting for such unmeasured variables. Therefore, we use an ADMG, which can represent latent variable-induced covariation and confounding through directed and bidirected edges. Consequently, in the presence of unmeasured variables, the problem reduces to estimating the structure of an ADMG. This section builds on the work of Bhattacharya et al. (2021).

Figure 1(a) depicts a DAG, which represents the relationships among the variables in the absence of unmeasured variables. Figures (b), (c), and (d) illustrate examples of ADMGs that we aim to estimate in this study. Figure (b) shows an ancestral ADMG, where no directed path $X_i \rightarrow \cdots \rightarrow X_j$ and bidirected edges $X_i \leftrightarrow X_j$ exist simultaneously in $G$ for any pair of vertices $X_i, X_j \in X$. Figure (c) shows an arid ADMG that does not contain any c-trees. A c-tree is a subgraph of $G$ where its directed edges form a directed tree, and its bidirected edges form a single bidirectional connected component within the subgraph. For details on c-trees, see Shpitser & Pearl (2006). Figure (d) shows a bow-free ADMGs, where no pair of vertices $X_i \rightarrow X_j$ and $X_i \leftrightarrow X_j$ both exist in $G$. These three types of ADMGs exhibit an inclusion relationship, with bow-free ADMGs being the most general type of ADMGs.

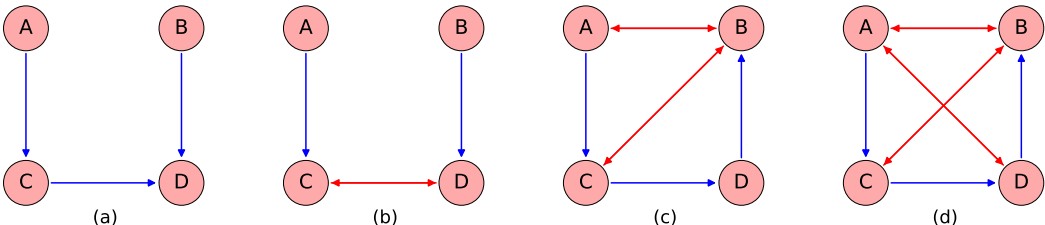

Figure 1: (a) DAG without unmeasured variables. (b) Ancestral ADMGs. (c) Arid ADMGs. (d) bow-free ADMGs.

$$\text{Ancestral} \subset \text{Arid} \subset \text{Bow-free}$$

In Bhattacharya et al. (2021), these ADMGs are expressed as differentiable constraints, allowing the selection of the appropriate ADMGs type to be estimated based on the data. This study adopts the differentiable constraints proposed by Bhattacharya et al. (2021), enabling the selection of suitable ADMGs types according to the data.

### 2.3 Identifiability in the Model

This section discusses the identifiability of the parameters in bow-free ADMGs where the error terms follow a multivariate generalized normal distribution. Brito & Pearl (2002) proved that given a bow-free ADMG model, the parameters are almost everywhere identifiable from the observed covariance matrix. Since this fact is often utilized under the assumption that the error terms are Gaussian, we show in this study that it also applies to bow-free ADMG models when the error terms follow a multivariate generalized normal distribution. Furthermore, we draw on Wang & Drton (2024), who demonstrated that the model can identify causal directions, and not only Markov equivalence classes, using the non-Gaussianity of error terms. This study also provides evidence that causal directions can be estimated.

#### 2.3.1 Definition and Key Terms

**bow-free ADMG.** A bow-free ADMG is a type of directed mixed graph (ADMG) that may include latent variables. For any two nodes $X_i$ and $X_j$, it disallows having both

$$X_i \to X_j \quad \text{and} \quad X_i \leftrightarrow X_j$$

simultaneously (i.e., no 'bow" shape). This ensures that any bidirected edge (representing covariance structures induced by latent variables) does not overlap with a directed edge between the same pair of nodes.

**Markov equivalence class.** A Markov equivalence class of DAGs (or ADMGs) is a set of graphs that encode the same set of conditional independence relationships. When the error terms are

assumed to be Gaussian, one often can only identify the Markov equivalence class, meaning that the directions of certain edges cannot be distinguished.

**Non-Gaussianity.** Non-Gaussianity refers to error terms that do not follow a Gaussian distribution (e.g., distributions with skewness or high kurtosis). Such non-Gaussian characteristics can allow one to distinguish causal directions that remain indistinguishable under purely Gaussian assumptions, by exploiting higher-order moments or skewness.

### 2.3.2 On the Identifiability of Parameters in bow-Free ADMGs with Multivariate Generalized Normal Distributions

Brito & Pearl (2002) demonstrated that a bow-free ADMG model is almost always identifiable from the observed covariance matrix. As the argument in Brito & Pearl (2002) primarily assumes that the error terms are Gaussian, we follow Brito & Pearl (2002) to demonstrate that when the error terms have a multivariate generalized normal distribution (MGGD), a bow-free ADMG model is almost always identifiable from the observed covariance matrix. (See the Appendix for the proof.)

**Theorem 1**

Let $G$ be a bow-free ADMG with error terms following a multivariate generalized normal distribution, and let the set of parameters of $G$ be $\theta = \{\delta, \Omega\}$. Then, for almost all $\theta$, the following holds:

$$\Sigma(\theta) = \Sigma(\theta')$$

implies

$$\theta = \theta'.$$

In other words, if two parameter sets $\theta$ and $\theta'$ yield the same covariance matrix $\Sigma$, then $\theta$ and $\theta'$ must be identical, except possibly when $\theta$ belongs to a set of Lebesgue measure zero.

This result follows from extending the argument of Brito & Pearl (2002) (which relies on Wright's path analysis and the inductive construction of linear SEMs) to the MGGD setting. Because the MGGD includes the Gaussian distribution (as a special case with $\beta = 1$) and is closed under linear transformations, imposing the structural constraints of a bow-free ADMG yields a unique parameter set $\{\delta, \Omega\}$.

Moreover, if the error terms are non-Gaussian (i.e., $\beta \neq 1$), then not only are the parameters identifiable, but there is also potential to determine the directions of causal edges, as suggested by Wang & Drton (2024) and others. In this paper, we leverage this non-Gaussianity to propose a method for estimating both the structure and orientations in bow-free ADMGs.

**Intuition.** Because the generalized normal distribution includes the Gaussian as a special case, each valid $(\delta, \Omega)$ pair generally induces a unique covariance structure, making parameters identifiable in the bow-free setting.

### 2.3.3 Identifiability of the structure in bow-Free ADMGs Using Non-Gaussianity

Wang & Drton (2024) demonstrated that when the data are non-Gaussian and correspond to a bow-free ADMG, the bow-free ADMG can be consistently recovered, including both the Markov equivalence class and causal directions. The multivariate generalized normal distribution assumed

for the observed data in this study is non-Gaussian except in the special case. Therefore, unlike Bhattacharya et al. (2021), who were limited to recovering the Markov equivalence class, it is expected that the causal directions can also be estimated.

## 3 Decomposition of the Log-Likelihood Function of the Multivariate Generalized Normal Distribution

### 3.1 Probability Density Function of the Multivariate Generalized Normal Distribution

As defined by Gómez et al. (1998), the probability density function of the multivariate generalized Gaussian distribution (MGGD) is given by:

$$f(X \mid \mu, \Sigma, \beta) = \frac{\Gamma\left(\frac{p}{2}\right)}{\pi^{\frac{p}{2}} \, \Gamma\left(\frac{p}{2\beta}\right)} \cdot \frac{\beta}{2^{\frac{p}{2\beta}} \, |\Sigma|^{\frac{1}{2}}} \exp\left(-\tfrac{1}{2}\left((X-\mu)^\top \Sigma^{-1}(X-\mu)\right)^\beta\right), \tag{4}$$

where $X$ is a $p$-dimensional random vector ($p \geq 1$) that follows a power-exponential distribution with parameters $\mu$, $\Sigma$, and $\beta$. Specifically, $\mu \in \mathbb{R}^p$, $\Sigma$ is a $(p \times p)$ positive-definite symmetric matrix, and $\beta \in (0, \infty)$. $\Gamma(\cdot)$ denotes the gamma function. Notably, the MGGD reduces to a multivariate normal distribution when $\beta = 1$. In this distribution, if $\Sigma$ is a diagonal matrix, then the correlation coefficients between the components become zero. However, because the multivariate generalized normal distribution belongs to the elliptical family, a zero correlation does not imply independence. Nonetheless, if the components are assumed to be generated independently, they can be considered truly independent rather than merely uncorrelated.

Gómez et al. (1998) show that the MGGD is invariant under affine transformations. More precisely, if $f(X \mid \mu_X, \Sigma_X, \beta)$ is the probability density function, then for the affine transformation

$$Y = CX + b, \tag{5}$$

where $C$ is a nonsingular matrix, $b$ is a vector in $\mathbb{R}^p$, and the transformed variable $Y$ follows $f(Y \mid C\mu_X + b, \, C\,\Sigma_X\,C^\top, \, \beta)$. This indicates that the family of distributions remains within the same class under any nonsingular linear transformation and translation.

This affine transformation property is particularly useful in linear structural equation models (especially (2)). In such models, the relationships among the observed variables and the covariance structure of the error terms are modeled, leading to the covariance matrix $\Sigma$ of the observed variables in the form

$$\Sigma = (I - \delta)^{-1}\,\Omega\,(I - \delta)^{-\top}.$$

Furthermore, as discussed in Section 3.2.1, one can estimate the parameters $\delta$ and $\Omega$ from the observed covariance matrix. In other words, if the data follow an MGGD, then in principle $\delta$ and $\Omega$ can be estimated from an observed covariance matrix of the form $\Sigma = (I - \delta)^{-1}\,\Omega\,(I - \delta)^{-\top}$.

**Key Point (6):** The elliptical nature and affine-invariance of the MGGD are crucial for applying it to linear SEMs. They ensure that under linear transformations (e.g. confounding), the distributional form remains within the same family, aiding our later decomposition.

### 3.2 Log-Likelihood Function

Assuming that in the ADMG graph $G = (V, E)$ of the linear model (2), $N$ observations are drawn, where all the components of $\mu$ are zero (the mean vector is zero). The reason for setting all the components of $\mu$ to zero is to prevent notational clutter without loss of generality. In this case, the log-likelihood function is given by (4) as follows:

$$
\begin{aligned}
\ell(\mu, \Sigma, \beta | X) = {} & N \log \Gamma \left(\frac{p}{2}\right) + N \log \beta - \frac{p}{2} N \log \pi \\
& - N \log \Gamma \left(\frac{p}{2\beta}\right) - \frac{p}{2\beta} N \log 2 - \frac{N}{2} \log |\Sigma| \\
& - \frac{1}{2} \sum_{l=1}^{N} \left(X^{(l)\top} \Sigma^{-1} X^{(l)}\right)^{\beta}
\end{aligned}
\tag{6}
$$

### 3.3 Decomposition of the Log-Likelihood Function

The main component of the proposed algorithm is the decomposition of the log-likelihood function of the multivariate generalized normal distribution, inspired by Dorton et al. (2009), who decomposed the log-likelihood function of the multivariate normal distribution.

Let $X_i \in \mathbb{R}^N$ denote the $i$th row of the observation matrix $X$ and $X_{-i} = X_{V/i}$ be the $(V \setminus \{i\}) \times N$ submatrix of $X$. We adopt the abbreviated notation $X_C$ to represent the $C \times N$ submatrix of the $D \times N$ matrix $X$, where $C \leq D$.

**Theorem 3**

Let $i \in X$ be a variable node in the ADMG graph $G = (V, E)$ of the linear model (2). Let $\|x\|^2 = x^\top x$ and define $\Omega_{ii.-i}$ as the conditional variance of $\varepsilon_i$ given $\varepsilon_{-i}$ as follows:

$$
\Omega_{ii.-i} = \Omega_{ii} - \Omega_{i,-i} \Omega_{-i,-i}^{-1} \Omega_{-i,i}
\tag{7}
$$

Here, $\Omega_{i,-i}$ is the row vector obtained by removing the $i$th element from the $i$th row, $\Omega_{-i,i}$ is the column vector obtained by removing the $i$th element from the $i$th column, and $\Omega_{-i,-i}$ denotes the submatrix obtained by removing the $i$th row and $i$th column from $\Omega$. Additionally, let $\Omega_{-i,-i}^{-1} = (\Omega_{-i,-i})^{-1}$. Then, the log-likelihood function $\ell(B, \Omega, \beta)$ of the graph $G = (V, E)$ can be decomposed as

$$
\begin{aligned}
\ell(\mu, \Sigma, \beta \mid X) = {} & -\frac{N}{2} \log \Omega_{ii.-i} - \frac{N}{2} \log \det(\Omega_{-i,-i}) \\
& - \frac{1}{2} \sum_{l=1}^{N} \left( \Omega_{ii.-i}^{-1} \left( \left(X_i^{(l)} - \delta_{i,\mathrm{pa}(i)} X_{\mathrm{pa}(i)}^{(l)} - \Omega_{i,\mathrm{sp}(i)} \left(\Omega_{-i,-i}^{-1} \varepsilon_{-i}^{(l)}\right)_{\mathrm{sp}(i)}\right)^2 \right.\right. \\
& \hphantom{- \frac{1}{2} \sum_{l=1}^{N} \Big(} \left.\left. + \varepsilon_{-i}^{(l)\top} \Omega_{-i,-i}^{-1} \varepsilon_{-i}^{(l)} \right) \right)^{\beta}.
\end{aligned}
\tag{8}
$$

*Proof.* By rearranging the log-likelihood function described in equation (6), considering the constant parts unrelated to the coefficient matrices $\delta$ and $\Omega$, noting that the determinant $\det(I - \delta) = 1$

since $\delta$ is acyclic, and that $(I - \delta)X = \varepsilon$, and considering that the covariance matrix $\Sigma$ can be expressed using the adjacency matrix $B$ and $\Omega$ as $\Sigma = \mathrm{Var}(X) := (I - \delta)^{-1}\Omega(I - \delta)^{-\top}$, we obtain the following:

$$
\begin{aligned}
\ell(\mu, \Sigma, \beta | X) &= -\frac{N}{2} \log |\Sigma| - \frac{1}{2} \sum_{l=1}^{N} \left( X^{(l)\top} \Sigma^{-1} X^{(l)} \right)^{\beta} \\
&= \frac{N}{2} \log |(I - \delta)\Omega(I - \delta)^{\top}| - \frac{1}{2} \sum_{l=1}^{N} \left( X^{(l)\top}(I - \delta)^{\top}\Omega^{-1}(I - \delta)X^{(l)} \right)^{\beta} \\
&= -\frac{N}{2} \log |\Omega| - \frac{1}{2} \sum_{l=1}^{N} \left( \varepsilon^{(l)\top}\Omega^{-1}\varepsilon^{(l)} \right)^{\beta}
\end{aligned}
\tag{9}
$$

We can partition $\Omega$ as a block matrix:

$$
\Omega = \begin{pmatrix} \Omega_{ii} & \Omega_{i,-i} \\ \Omega_{-i,i} & \Omega_{-i,-i} \end{pmatrix}
\tag{10}
$$

Based on equations (7) and (10), we can rearrange $\log |\Omega|$ as shown in equation (11):

$$
\begin{aligned}
\log |\Omega| &= \log \left( \Omega_{ii} - \Omega_{i,-i}\Omega_{-i,-i}^{-1}\Omega_{-i,i} \right) + \log |\Omega_{-i,-i}| \\
&= \log \Omega_{ii.-i} + \log |\Omega_{-i,-i}|
\end{aligned}
\tag{11}
$$

Next, we rearrange the term $\varepsilon^{(l)\top}\Omega^{-1}\varepsilon^{(l)}$ in $\frac{1}{2}\sum_{l=1}^{N}\left(\varepsilon^{(l)\top}\Omega^{-1}\varepsilon^{(l)}\right)^{\beta}$. We partition $\Omega^{-1}$ as a block matrix:

$$
\Omega^{-1} = \begin{pmatrix} \Omega_{ii} & \Omega_{i,-i} \\ \Omega_{-i,i} & \Omega_{-i,-i} \end{pmatrix}^{-1} = \begin{pmatrix} \Omega_{ii.-i}^{-1} & -\Omega_{ii.-i}^{-1}\Omega_{i,-i}\Omega_{-i,-i}^{-1} \\ -\Omega_{-i,-i}^{-1}\Omega_{-i,i}\Omega_{ii.-i}^{-1} & \Omega_{-i,-i}^{-1} + \begin{array}{c}\Omega_{-i,-i}^{-1}\Omega_{-i,i}\Omega_{ii.-i}^{-1} \\ \cdot\,\Omega_{i,-i}\Omega_{-i,-i}^{-1}\end{array} \end{pmatrix}.
\tag{12}
$$

Considering that $\Omega^{-1}$ is a block matrix, we can rearrange $\varepsilon^{(l)\top}\Omega^{-1}\varepsilon^{(l)}$ as follows:

$$
\begin{aligned}
\varepsilon^{(l)\top}\Omega^{-1}\varepsilon^{(l)} &= \begin{pmatrix} \varepsilon_i^{(l)} & \varepsilon_{-i}^{(l)\top} \end{pmatrix} \begin{pmatrix} \Omega_{ii.-i}^{-1} & -\Omega_{ii.-i}^{-1}\Omega_{i,-i}\Omega_{-i,-i}^{-1} \\ -\Omega_{-i,-i}^{-1}\Omega_{-i,i}\Omega_{ii.-i}^{-1} & \Omega_{-i,-i}^{-1} + \Omega_{-i,-i}^{-1}\Omega_{-i,i}\Omega_{ii.-i}^{-1}\Omega_{i,-i}\Omega_{-i,-i}^{-1} \end{pmatrix} \begin{pmatrix} \varepsilon_i^{(l)} \\ \varepsilon_{-i}^{(l)} \end{pmatrix} \\
&= \Omega_{ii.-i}^{-1}(\varepsilon_i^{(l)} - \Omega_{i,-i}\Omega_{-i,-i}^{-1}\varepsilon_{-i}^{(l)})^2 + \varepsilon_{-i}^{(l)\top}\Omega_{-i,-i}^{-1}\varepsilon_{-i}^{(l)}
\end{aligned}
\tag{13}
$$

From this rearrangement, the log-likelihood function becomes equation (14):

$$
\begin{aligned}
\ell(\mu, \Sigma, \beta \mid X) = &-\frac{N}{2} \log \Omega_{ii.-i} - \frac{N}{2} \log \det(\Omega_{-i,-i}) \\
&+ \frac{1}{2} \sum_{l=1}^{N} \left( \Omega_{ii.-i}^{-1}\left( (\varepsilon_i^{(l)} - \Omega_{i,-i}\Omega_{-i,-i}^{-1}\varepsilon_{-i}^{(l)})^2 + \varepsilon_{-i}^{(l)\top}\Omega_{-i,-i}^{-1}\varepsilon_{-i}^{(l)} \right) \right)^{\beta}.
\end{aligned}
\tag{14}
$$

By definition, the error term $\varepsilon_i^{(l)} = X_i^{(l)} - \delta_{i,\mathrm{pa}(i)} X_{\mathrm{pa}(i)}^{(l)}$. Moreover, since we are dealing with bow-free ADMGs, we have $\Omega_{i,-i} \Omega_{-i,-i}^{-1} \varepsilon_{-i}^{(l)} = \Omega_{i,\mathrm{sp}(i)} \left( \Omega_{-i,-i}^{-1} \varepsilon_{-i}^{(l)} \right)_{\mathrm{sp}(i)}$. This yields the claimed decomposition.

$\square$

The decomposition of the log-likelihood function is based on decomposing the joint distribution of $\varepsilon$ into the marginal distribution of $\varepsilon_{-i}$ and conditional distribution $(\varepsilon_i \mid \varepsilon_{-i})$. In particular, as shown in (14), the squared term $(\varepsilon_i^{(l)} - \Omega_{i,-i} \Omega_{-i,-i}^{-1} \varepsilon_{-i}^{(l)})^2$ represents the deviation of $\varepsilon_i$ from its conditional expectation given $\varepsilon_{-i}$, which plays a key role in deriving the likelihood decomposition. This idea leads to an approach similar to that of Dorton et al. (2009), who decomposed the log-likelihood function of the multivariate normal distribution and proposed an iterative algorithm. The steps of this algorithm are based on fixing the marginal distribution of $\varepsilon_{-i}$ and estimating the conditional distribution. To fix the marginal distribution of $\varepsilon_{-i}$, we must fix the submatrix of $\Omega_{-i,-i}$ excluding the $i$th row and $i$th column and the submatrix of $\delta_{-i,V}$ excluding the $i$th row. This is because $\varepsilon_{-i}$ is determined depending on $\Omega_{-i,-i}$ and $\delta_{-i,V}$.

**(Intuition.)** By isolating each node's contribution and conditioning out the rest, the log-likelihood can be decomposed into local pieces that depend on the parents and siblings (in terms of bidirected edges). Such an approach is crucial for efficiently optimizing over ADMG structures under the MGGD.

Before presenting the full decomposition, we first define the pseudo-variable $Z_{-i}$ as follows:

$$Z_{-i} = \Omega_{-i,-i}^{-1} \varepsilon_{-i}, \tag{15}$$

where $\Omega_{-i,-i}$ is the submatrix of $\Omega$ obtained by removing the $i$-th row and column, and $\varepsilon_{-i}$ is the error vector excluding the $i$-th component. We will use $Z_{-i}$ in the subsequent derivations to simplify the notation of the conditional distribution.

Note that $\Omega_{-i,-i}^{-1} \varepsilon_{-i}$ in (equation 15) captures the same quantity that appears in the decomposition of $\varepsilon_i$ given $\varepsilon_{-i}$. In later equations, if we write $\Omega_{i,-i} \, \Omega_{-i,-i}^{-1} \varepsilon_{-i}$, it should be understood that the vector part $\Omega_{-i,-i}^{-1} \varepsilon_{-i}$ is precisely $Z_{-i}$.

$\ell(\mu, \Sigma, \beta | X)$

$$= -\frac{N}{2} \log \Omega_{ii.-i} - \frac{1}{2} \sum_{l=1}^{N} \left( \Omega_{ii.-i}^{-1} \left( X_i^{(l)} - \delta_{i,\mathrm{pa}(i)} X_{\mathrm{pa}(i)}^{(l)} - \Omega_{i,\mathrm{sp}(i)} \left( \Omega_{-i,-i}^{-1} \varepsilon_{-i}^{(l)} \right)_{\mathrm{sp}(i)} \right)^2 \right)^{\beta}$$

$$= -\frac{N}{2} \log \Omega_{ii.-i} - \frac{1}{2\Omega_{ii.-i}^{\beta}} \sum_{l=1}^{N} \left( \left( X_i^{(l)} - \sum_{j \in \mathrm{pa}(i)} \delta_{i,j} X_j^{(l)} - \sum_{k \in sp(i)} \Omega_{i,k} Z_k^{(l)} \right)^2 \right)^{\beta} \tag{16}$$

Assuming $\beta \geq 1$, we rearrange equation (16) using Hölder's inequality(for details on the application of Hölder's inequality, see the APPENDIX).

$\ell(\mu, \Sigma, \beta | X)$

$$
= -\frac{N}{2} \log \Omega_{ii.-i} - \frac{1}{2\Omega_{ii.-i}^{\beta}} \sum_{l=1}^{N} \left( \left( X_i^{(l)} - \sum_{j \in \mathrm{pa}(i)} \delta_{i,j} X_j^{(l)} - \sum_{k \in sp(i)} \Omega_{i,k} Z_k^{(l)} \right)^2 \right)^{\beta}
$$

$$
= -\frac{N}{2} \log \Omega_{ii.-i} - \frac{1}{2\Omega_{ii.-i}^{\beta}} \frac{N^{\beta-1}}{N^{\beta-1}} \sum_{l=1}^{N} \left( \left( X_i^{(l)} - \sum_{j \in \mathrm{pa}(i)} \delta_{i,j} X_j^{(l)} - \sum_{k \in sp(i)} \Omega_{i,k} Z_k^{(l)} \right)^2 \right)^{\beta}
$$

$$
\geq -\frac{N}{2} \log \Omega_{ii.-i} - \frac{1}{2\Omega_{ii.-i}^{\beta}} \frac{1}{N^{\beta-1}} \left( \sum_{l=1}^{N} \left( X_i^{(l)} - \sum_{j \in \mathrm{pa}(i)} \delta_{i,j} X_j^{(l)} - \sum_{k \in sp(i)} \Omega_{i,k} Z_k^{(l)} \right)^2 \right)^{\beta}
$$

$$
= -\frac{N}{2} \log \Omega_{ii.-i} - \frac{1}{2\Omega_{ii.-i}^{\beta}} \frac{1}{N^{\beta-1}} \left\| X_i - \sum_{j \in \mathrm{pa}(i)} \delta_{i,j} X_j - \sum_{k \in sp(i)} \Omega_{i,k} Z_k \right\|^{2\beta}
$$

$$\tag{17}$$

Maximizing equation (17) to estimate $\delta$ and $\Omega$ is equivalent to performing a regression of $X_i$ (as the target variable) on $X_j$ (the parent variable of $X_i$) and the pseudo-variables $Z_k$, considering the shape parameter $\beta$.

Utilizing these observations, in the next section, we propose a method for causal structure estimation using continuous optimization, considering the presence of unmeasured variables and assuming that the error variables follow a multivariate generalized normal distribution, using the decomposition results of the log-likelihood function organized in this section to estimate $\hat{\delta}, \hat{\Omega}$.

## 4  Proposed Method

### 4.1  Causal Discovery Based on Differentiable Scores

Score-based methods aim to estimate causal structures by maximizing a graph's score (e.g., log-likelihood) given the data. Learning DAGs from data is an NP-hard problem because it is challenging to efficiently enforce combinatorial acyclicity constraints ((Chickering, 1996)).

Zheng et al. (2018) proposed a new approach for score-based DAG learning by converting the traditional combinatorial optimization problem (18) into a continuous optimization problem (19).

$$
\min_{\theta \in \Theta} F(\theta) \quad \text{subject to } G(\theta) \in \text{DAGs} \tag{18}
$$

$$
\min_{\theta \in \Theta} F(\theta) \quad \text{subject to } h(\theta) = 0, \tag{19}
$$

where $G(\theta)$ is a $d$-node graph induced by the weight matrix $\theta \in \mathbb{R}^{d \times d}$, and $F : \mathbb{R}^{d \times d} \to \mathbb{R}$ is a score function. $h : \mathbb{R}^{d \times d} \to \mathbb{R}$ is a smooth function over real matrices, and the constraint $h(\theta) = 0$

can precisely characterize the acyclicity of the graph. Causal structure estimation via continuous optimization eliminates the need for specialized algorithms to explore the combinatorial space of DAGs and instead allows the use of standard numerical algorithms for constrained problems, making implementation particularly straightforward, as mentioned in Zheng et al. (2018).

The acyclicity constraint is defined as follows:

$$h(\theta) = \text{trace}\left(e^{\theta \circ \theta}\right) - d = 0 \tag{20}$$

Here, $\theta \circ \theta$ denotes the Hadamard product (element-wise multiplication), trace $\left(e^{\theta \circ \theta}\right)$ is the trace (sum of the diagonal elements) of the matrix exponential, and $d$ is the number of variables. This constraint ensures that the matrix $\theta$ forms a DAG.

Zheng et al. (2018) used the augmented Lagrangian method as a continuous optimization technique. This method solves constrained optimization problems using an objective function that includes penalty terms and is formulated as

$$\min_{\theta \in \Theta} L(\theta) + \lambda \|\theta\|_1 + \alpha h(\theta) + \frac{\rho}{2} h(\theta)^2 \quad \text{subject to} \quad h(\theta) = 0. \tag{21}$$

Here, $\lambda$ is the weight of the regularization term, $\alpha$ is the Lagrange multiplier, and $\rho$ is the penalty coefficient.

**Key Point (7):** Score-based methods combined with *differentiable* constraints allow us to leverage standard optimization routines (e.g. gradient-based) instead of combinatorial searches, thus scaling more easily to larger graphs.

## 4.2 ABIC

Although the differentiable score-based causal discovery method using continuous optimization proposed by Zheng et al. (2018) has been successful in estimating causal structures in DAGs, it cannot be directly applied to ADMGs. This is because ADMGs require two adjacency matrices, $D$ and $B$, to represent the directed and bidirected edges, respectively. To extend the differentiable algebraic characterization to ADMGs, Bhattacharya et al. (2021) proposed differentiable constraints for each causal graph. In Bhattacharya et al. (2021), three differentiable constraints (i.e., ancestral, arid, and bow-free)are proposed for ADMGs , as shown in Table 1.

| ADMGs | Algebraic Constraint |
|---|---|
| Ancestral | $\text{trace}(e^D) - d + \text{sum}(e^D \circ B) = 0$ |
| Arid | $\text{trace}(e^D) - d + \text{Greenery}(D, B) = 0$ |
| Bow-free | $\text{trace}(e^D) - d + \text{sum}(D \circ B) = 0$ |

Table 1: Differentiable constraints for each causal graph in ADMGs (Bhattacharya et al., 2021).

For example, to estimate the causal structure of bow-free ADMGs, the constraint equation becomes (22).

$$h(\theta) = \operatorname{tr}(e^D) - d + \operatorname{sum}(D \circ B) \tag{22}$$

Here, $\operatorname{tr}(A)$ denotes the trace of the square matrix $A$, i.e., the sum of its diagonal entries.

Here, $\operatorname{sum}(\cdot)$ denotes the sum of all the elements in a matrix. It has been proven that when $h(\theta) = 0$, the estimated graph corresponds to an ADMG ((Bhattacharya et al., 2021)). Essentially, $\operatorname{tr}(e^D) - d$ signifies the standard acyclicity constraint for directed edges, and the latter term $\operatorname{sum}(D \circ B)$ ensures that bidirected edges are not introduced when a directed edge exists (i.e., it enforces the bow-free ADMGs property).

Bhattacharya et al. (2021) used the augmented Lagrangian method, similar to Zheng et al. (2018), to convert the problem into an optimization problem with a quadratic penalty term, and proposed ABIC, which solves the following primal equation at each iteration:

$$\min_{\theta \in \Theta} ABIC_\lambda(X; \theta) + \frac{\rho}{2} |h(\theta)|^2 + \alpha h(\theta), \tag{23}$$

where $\rho$ is the weight of the penalty term, and $\alpha$ is the Lagrange multiplier. Then, the Lagrange multiplier is updated as $\alpha \leftarrow \alpha + \rho h(\theta)$. Intuitively, by optimizing the primal equation with a large $\rho$, we force $h(\theta)$ to be very close to zero, thus satisfying the equality constraint.

**(Intuition.)** By formulating ADMG constraints as smooth penalty terms, we can directly incorporate them into standard optimization frameworks, rather than enumerating all possible graphs. This approach generalizes well beyond DAGs to bow-free or ancestral ADMGs.

### 4.3 ABIC LiNGAM

In this study, we extend the method proposed by Bhattacharya et al. (2021) and present algorithm2 that can estimate causal structures when the error terms follow a multivariate generalized normal distribution. The basic framework is the same as that in Bhattacharya et al. (2021), but we consider that the error terms follow a multivariate generalized normal distribution and incorporate the shape parameter $\beta$ into the loss function. The shape parameter must be estimated from the observed data in advance. We use the estimated shape parameter $\hat{\beta}$. In addition, since the multivariate normal distribution corresponds to $\beta = 1$, the proposed method can handle the normal distribution case, thereby generalizing the method of Bhattacharya et al. (2021). Depending on the data, it is possible to switch between non-Gaussianity and Gaussianity.

**Key Point (8):** By generalizing from ABIC (Gaussian) to ABIC LiNGAM (non-Gaussian), we can exploit higher-order statistics to potentially identify the orientations of edges in ADMGs. This is especially powerful when hidden confounders introduce bidirected edges.

**Comparison of ABIC vs. ABIC LiNGAM:**

1. **Residual norm:**
   - ABIC uses $\| \cdot \|^2$ (Gaussian assumption).
   - ABIC LiNGAM uses $\| \cdot \|^{2\beta}$ for non-Gaussian errors if $\beta \neq 1$.

---

**Algorithm 1:** Algorithm 1: ABIC

---

**Input** : $X \in \mathbb{R}^{n \times d}$ : Observed data
$\Omega \in \mathbb{R}^{d \times d}$ : Initial error covariance
$tol > 0$ : Convergence threshold
$max\_iters$ : Max iterations
$h(\theta)$ : Differentiable constraint (e.g. bow-free)
$\rho, \alpha$ : Augmented Lagrangian coefficients
$\lambda$ : Regularization weight

**Output:** $\delta^t, \Omega^t$ : Final estimates

**Initialization:**

1. $\delta^0, \Omega^0 \leftarrow (0, I_d)$ or random.
2. $c \leftarrow \ln(n)$ (used in $\tanh(c|\theta|)$).
3. Define

$$LS(\theta) = \frac{1}{2n} \sum_{i=1}^{d} \| X_{\cdot,i} - X \delta_{\cdot,i} - Z^{(i)} \Omega_{\cdot,i} \|^2.$$

*(Here, $Z^{(i)}$ are pseudo-variables updated per iteration.)*

**for** $t = 0$ **to** $max\_iters - 1$ **do**

  **(A) Pseudo-variables:**

    **for** $i = 1$ **to** $d$ **do**

      $\epsilon_i \leftarrow X_{\cdot,i} - X \delta_{\cdot,i}^t$.

      $Z_{\cdot,i}^{(i)} \leftarrow 0, \quad Z_{\cdot,-i}^{(i)} \leftarrow \epsilon_{-i} (\Omega_{-i,-i}^t)^{-T}$.

  **(B) Parameter optimization:**

    Solve

$$(\delta^{t+1}, \Omega^{t+1}) \leftarrow \arg\min_\theta \Big\{ LS(\theta) + \frac{\rho}{2}|h(\theta)|^2 + \alpha\, h(\theta) + \lambda \sum_j \tanh\big(c|\theta_j|\big) \Big\}.$$

    **for** $i = 1$ **to** $d$ **do**

      $\epsilon_i \leftarrow X_{\cdot,i} - X \delta_{\cdot,i}^{t+1}$.

      $\Omega_{ii}^{t+1} \leftarrow \mathrm{var}(\epsilon_i)$.

  **(C) Convergence check:**

    **if** $\|\delta^{t+1} - \delta^t\| + \|\Omega^{t+1} - \Omega^t\| < tol$ **then**

      **break**

**return** $\delta^t, \Omega^t$

**Tips:**

- $h(\theta) = 0$ can enforce a DAG or ADMG constraint (e.g. bow-free).
- Increase $\rho$ gradually if $h(\theta)$ stays large.
- If $\delta^0 = 0$, $\Omega^0 = I_d$, residuals start simply.

---

2. **Shape parameter $\beta$:**

- ABIC LiNGAM requires $\beta > 0$.
- If $\beta = 1$, it becomes exactly the Gaussian case (Algorithm 1).

---

**Algorithm 2:** Algorithm 2: ABIC LiNGAM

---

**Input** : $X \in \mathbb{R}^{n \times d}$ : Observed data

$\Omega \in \mathbb{R}^{d \times d}$ : Initial error covariance

$tol > 0$ : Convergence threshold

$max\_iters$ : Max iterations

$h(\theta)$ : Differentiable constraint

$\rho, \alpha$ : Augmented Lagrangian coeff.

$\lambda$ : Regularization weight

$\beta > 0$ : Shape parameter (if $\beta \neq 1$, non-Gaussian)

**Output:** $\delta^t, \Omega^t$

**Initialization:**

1. $\delta^0, \Omega^0 \leftarrow (0, I_d)$ or random.
2. $c \leftarrow \ln(n)$.
3. Define

$$LS(\theta) = \frac{1}{2n} \sum_{i=1}^{d} \left\| X_{\cdot,i} - X \delta_{\cdot,i} - Z^{(i)} \Omega_{\cdot,i} \right\|^{2\beta}.$$

*(Note: $\beta = 1$ recovers the Gaussian case.)*

---

**for** $t = 0$ **to** $max\_iters - 1$ **do**

  **(A) Pseudo-variables:**

    **for** $i = 1$ **to** $d$ **do**

      $\epsilon_i \leftarrow X_{\cdot,i} - X \delta_{\cdot,i}^t.$

      $Z_{\cdot,i}^{(i)} \leftarrow 0, \quad Z_{\cdot,-i}^{(i)} \leftarrow \epsilon_{-i} \, (\Omega_{-i,-i}^t)^{-T}.$

  **(B) Parameter optimization:**

    Solve

$$(\delta^{t+1}, \Omega^{t+1}) \leftarrow \arg\min_\theta \left\{ LS(\theta) + \tfrac{\rho}{2} |h(\theta)|^2 + \alpha \, h(\theta) + \lambda \sum_j \tanh\!\big(c \, |\theta_j|\big) \right\}.$$

    **for** $i = 1$ **to** $d$ **do**

      $\epsilon_i \leftarrow X_{\cdot,i} - X \delta_{\cdot,i}^{t+1}.$

      $\Omega_{ii}^{t+1} \leftarrow \mathrm{var}(\epsilon_i).$

  **(C) Convergence check:**

    **if** $\|\delta^{t+1} - \delta^t\| + \|\Omega^{t+1} - \Omega^t\| < tol$ **then**

      **break**

**return** $\delta^t, \Omega^t$

**Tips:**

- If $\beta \neq 1$, non-Gaussian stats help identify directionality strictly.
- If $\beta = 1$, same as Algorithm 1.
- Use gradient-based methods or coordinate descent for the inner optimization.

---

3. **Identifiability: Orientation Recovery Beyond Gaussian Equivalence:**

- Under non-Gaussian assumptions ($\beta \neq 1$), ABIC LiNGAM can potentially identify causal directions.
- Gaussian ABIC recovers structure up to Markov equivalence.

## 5 Experiments

### 5.1 Simulation

#### 5.1.1 Simulation Setup and Evaluation

We aim to evaluate whether our proposed method (ABIC LiNGAM) can accurately recover bow-free ADMG structures under various conditions, including non-Gaussian error terms. Specifically, we focus on:

- How well the method identifies both the *skeleton* (presence or absence of edges) and the *direction* (arrowhead/tail) of each edge.

- The effect of different shape parameters $\beta$ of the multivariate generalized normal distribution (MGGD), which control the deviation from Gaussianity.

Following and extending the setup of Bhattacharya et al. (2021), we generate synthetic data from a *bow-free* ADMG as follows:

1. **Node pairs and edge assignment.** For each pair of nodes $(i, j)$ with $i < j$, draw a uniform random value in $[0, 1]$.

   - If this value is below a predefined threshold for *directed* edges, set $X_i \rightarrow X_j$ and sample the coefficient $\delta_{ij}$ uniformly in $[-2.0, -0.5] \cup [0.5, 2.0]$.
   - If this value is within the threshold for *bidirected* edges, set $X_i \leftrightarrow X_j$ and assign $\Omega_{ij} = \Omega_{ji}$ uniformly from $[-0.7, -0.4] \cup [0.4, 0.7]$.
   - Otherwise, no edge is placed between $(i, j)$.

2. **Diagonal entries of $\Omega$.** Each $\Omega_{ii}$ is sampled from an interval $\pm[0.4, 0.7]$. To ensure $\Omega$ is positive-definite, we add an adjustment term proportional to $\sum(|\Omega_{i,-i}|)$ plus an offset in $[0.1, 0.5]$.

3. **Shape parameter and error terms.** Let $\beta \in \{1, 3, 5\}$ be the shape parameter of the MGGD. For each node $i$, generate error terms $\epsilon_i$ from a multivariate generalized normal distribution (mean 0, covariance $\Omega$, shape $\beta$). Note that $\beta = 1$ recovers the Gaussian case.

4. **Generate observed data.** Construct $X_i$ via

$$X_i = \sum_{j \in \mathrm{pa}(i)} \delta_{ij} X_j + \epsilon_i, \quad i = 1, \ldots, d.$$

Repeating this for $n$ samples yields the data matrix $X \in \mathbb{R}^{n \times d}$.

This probabilistic framework ensures that the randomly generated $\delta$ and $\Omega$ conform to a bow-free ADMG while allowing flexible edge structures.

We vary:

- **Sample size** $n \in \{100, 500, 1000\}$,

- **Number of variables** $d \in \{5, 10\}$,

- **Shape parameter** $\beta \in \{1, 3, 5\}$ (MGGD).

Hence, we obtain $3 \times 2 \times 3 = 18$ total conditions. For each condition, we run 50 trials. All simulations use Python 3.8 with NumPy/SciPy. We used the hyperparameters recommended by ABIC (https://gitlab.com/rbhatta8/dcd) as a reference when setting up our implementation.

**Compared Methods.** We compare the proposed *ABIC LiNGAM* to:

- **bow-free ABIC** (Bhattacharya et al., 2021), which assumes Gaussian errors ($\beta = 1$) and recovers structures up to Markov equivalence.

- **FCI** (Spirtes et al., 2000), a constraint-based algorithm for ADMGs, outputting a partial ancestral graph (PAG).

- **BANG** (Wang & Drton, 2024), a constraint-based method that exploits higher-order moments to identify bow-free ADMG directions (5% significance).

Because our synthetic data are bow-free, we evaluate *ABIC LiNGAM* under the bow-free constraint.

We assess:

- *Skeleton accuracy* (presence or absence of edges): Precision, Recall, F1-score.

- *Arrowhead accuracy* (correctly oriented edges): Precision, Recall, F1-score.

- *Tail accuracy* (the tail end of directed edges): Precision, Recall, F1-score.

For FCI, which outputs a partial ancestral graph (PAG), we interpret $A \circ\!\!\rightarrow B$ as $A \rightarrow B$ and $\circ\!\!-\!\!\circ$ as $A \leftrightarrow B$.

**Key Point (9):** By introducing non-Gaussian errors (when $\beta \neq 1$), we can potentially distinguish causal directions beyond what is possible in purely Gaussian scenarios. Specifically, ABIC LiNGAM leverages score-based continuous optimization under a multivariate generalized normal assumption, providing the possibility of identifying orientations rather than merely recovering Markov equivalence classes.

### 5.1.2 Simulation Results and Discussion

In what follows, we report the performance of **ABIC LiNGAM** in terms of Recall, Precision, and F1-score on three aspects of the learned graph: *Skeleton*, *Arrowhead*, and *Tail*. We compare our

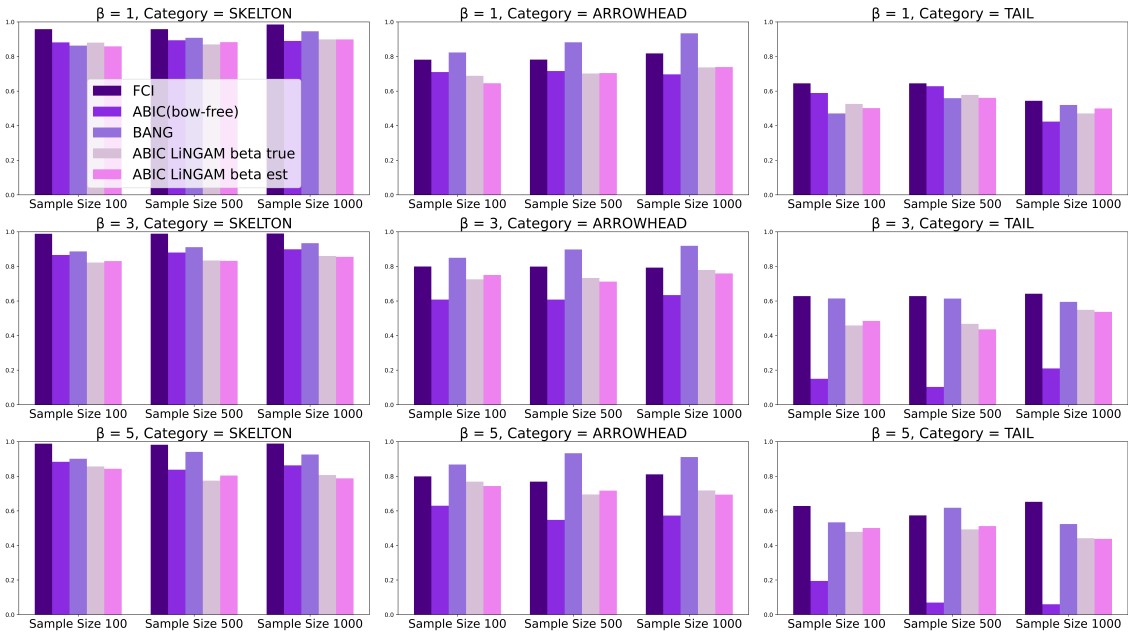

Figure 2: The precision results for each method with five variables.

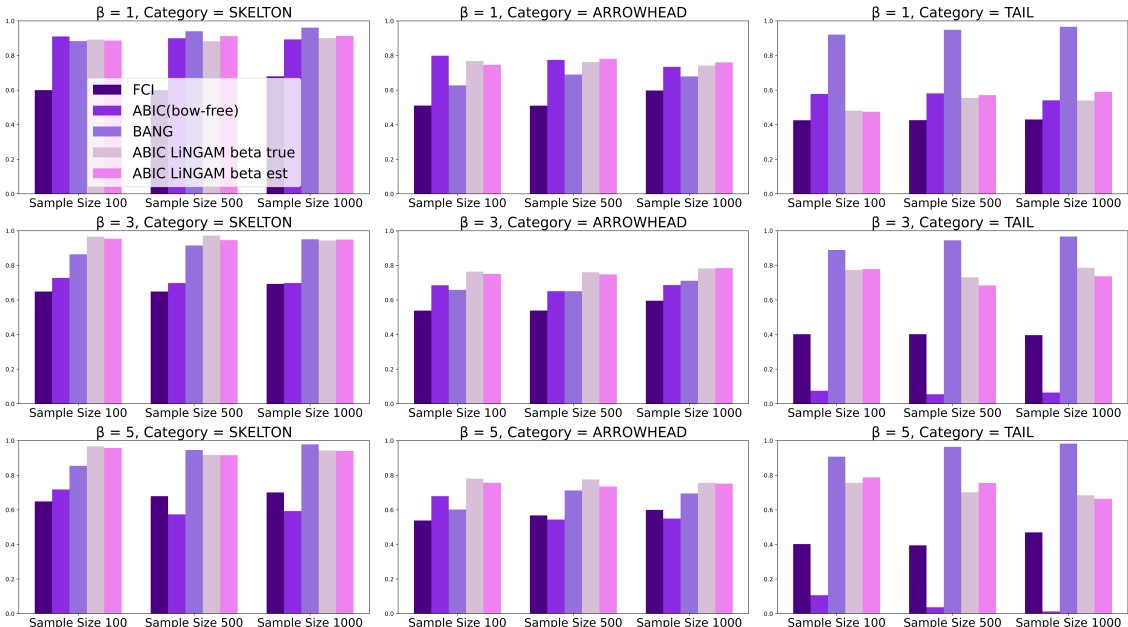

Figure 3: The recall results for each method with five variables.

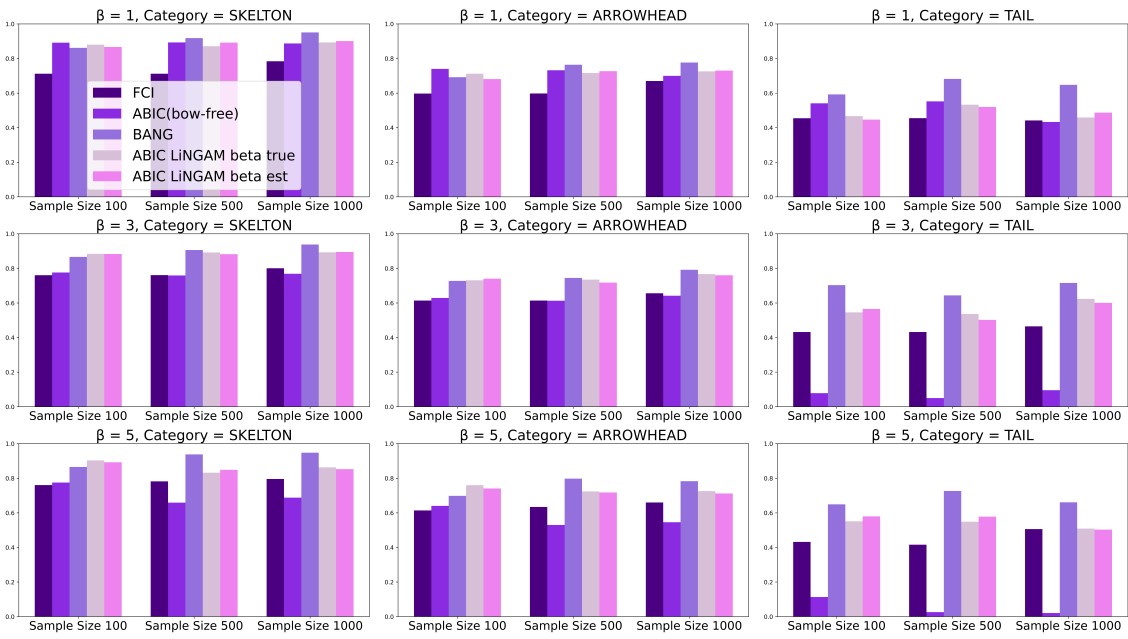

Figure 4: The F1-score results for each method with five variables.

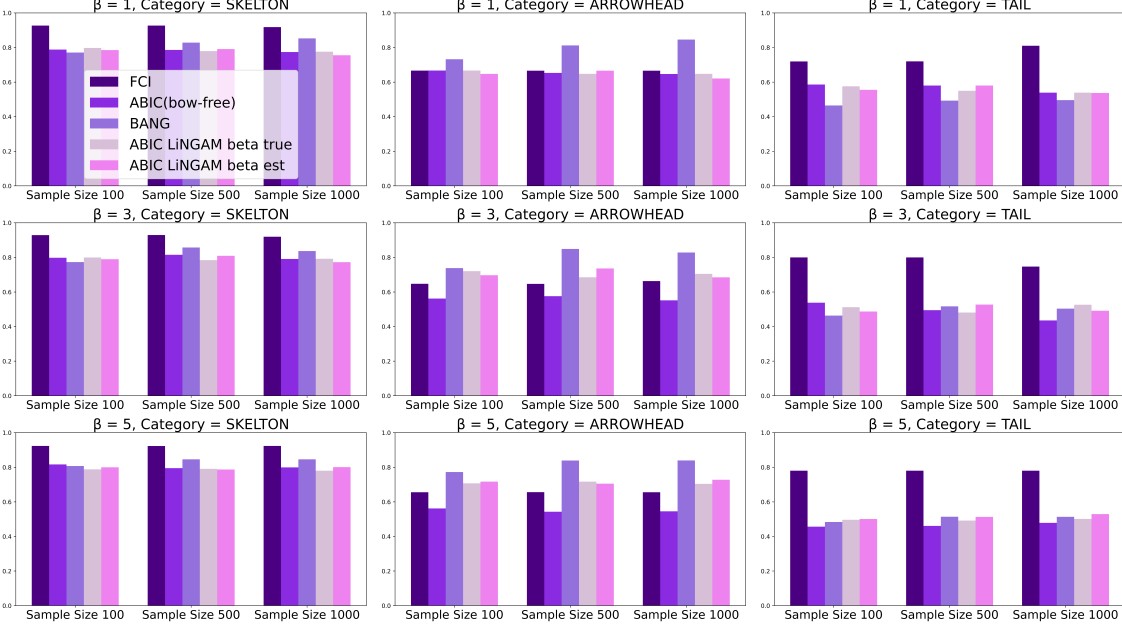

Figure 5: The precision results for each method with ten variables.

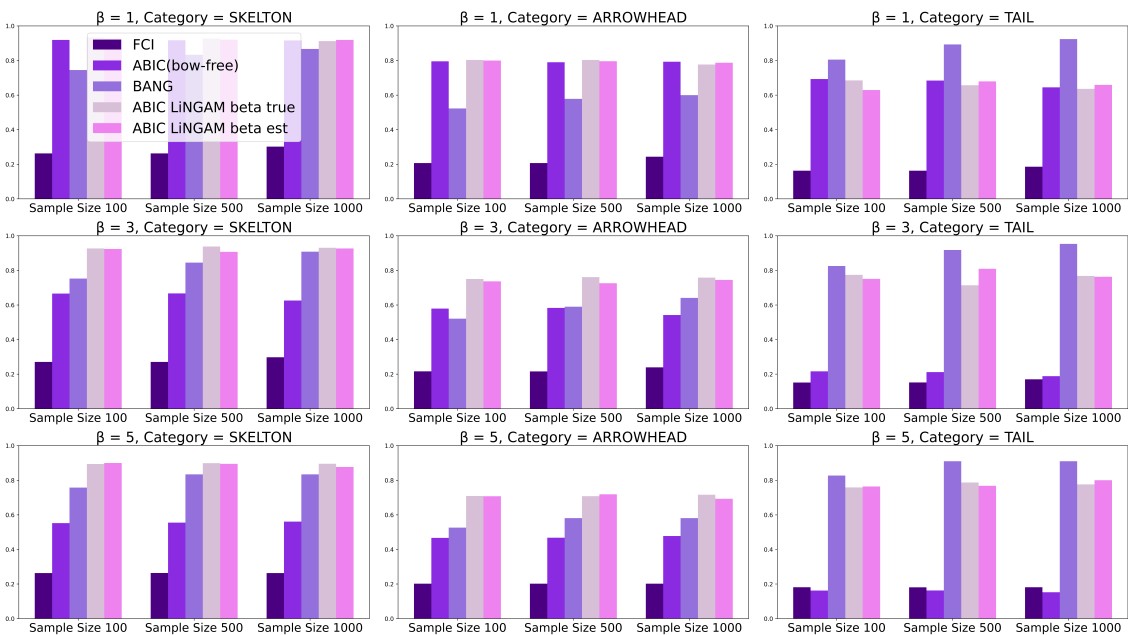

Figure 6: The recall results for each method with ten variables.

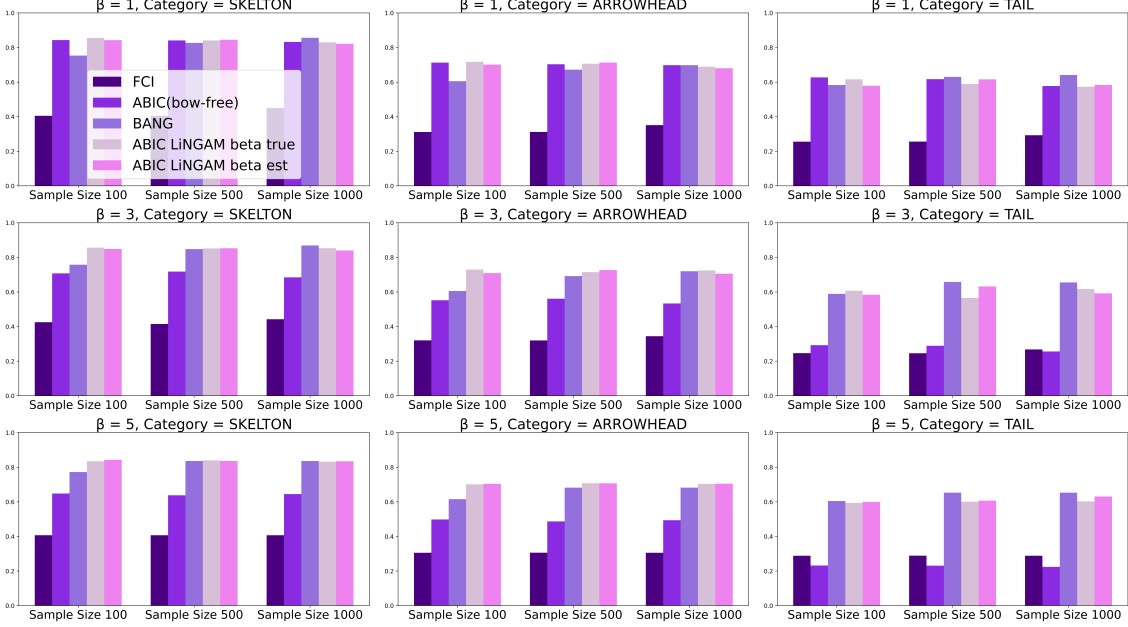

Figure 7: The F1-score results for each method with ten variables.

method with **BANG** (Wang & Drton, 2024), as well as ABIC (Bhattacharya et al., 2021) and FCI (Spirtes et al., 2000), under various conditions:

$$n \in \{100, 500, 1000\}, \quad d \in \{5, 10\}, \quad \beta \in \{1, 3, 5\}.$$

Key numerical results are shown in Figures (2)–(7).

**(1) Gaussian case:** $\beta = 1$.

- **Skeleton accuracy.** ABIC LiNGAM achieves almost the same Skeleton Recall/Precision/F1 as ABIC. This agrees with the fact that, in the Gaussian regime, both methods can recover structures only up to the Markov equivalence class.

- **Directional accuracy (Arrowhead/Tail).** Since $\beta = 1$ implies normal errors, ABIC LiNGAM does not gain additional information to identify directions uniquely. For instance, at $(n, d) = (500, 5)$, both methods yield high Skeleton-F1 but yield only limited Arrowhead/Tail correctness, consistent with standard LiNGAM theory.

**(2) Non-Gaussian case:** $\beta \neq 1$.

- **Improved direction recovery.** Once $\beta$ diverges from 1, ABIC LiNGAM can exploit non-Gaussianity to break Markov equivalences. For example, at $(d, n, \beta) = (5, 500, 3)$, ABIC LiNGAM obtains Arrowhead Recall 0.760, Precision 0.733, F1 0.735, surpassing BANG's Arrowhead Recall of 0.651 (though BANG reaches a high Precision of 0.898).

- **Consistency with increasing $\beta$.** We also observe stable performance as $\beta$ grows further, e.g., $(d, n, \beta) = (10, 500, 5)$ yields Arrowhead F1 $\approx 0.707$, close to the $\beta = 3$ case. This suggests that the proposed method's direction estimation remains robust across different levels of non-Gaussianity.

**(3) Effect of dimensionality ($d = 5$ vs. $d = 10$).**

- **Stability of ABIC LiNGAM at higher $d$.** Comparing $d = 5$ and $d = 10$ at $n = 500$, we see little degradation in ABIC LiNGAM's Arrowhead Recall and F1 (e.g., 0.726 vs. 0.760). Hence, the method appears well-suited for moderately larger graphs.

- **Comparison to BANG at $d = 10$.** As $d$ increases, the gap in directional accuracy between ABIC LiNGAM and BANG tends to narrow. For instance, at $(d, n, \beta) = (10, 500, 3)$, ABIC LiNGAM's Arrowhead Recall (0.726) exceeds BANG's (0.590), though BANG still sometimes shows a slightly better Tail precision in certain settings.

**(4) Overall comparison to existing methods.**

- **BANG** generally exhibits high directional accuracy, especially for $d = 5$. In particular, with Tail metrics at $(d, n, \beta) = (5, 500, 3)$, BANG shows Recall of 0.945, higher than ABIC LiNGAM's 0.684. That said, the difference decreases as $d$ grows.

- **ABIC** (Gaussian-only) matches ABIC LiNGAM under $\beta = 1$, but under non-Gaussianity, it struggles to recover directions (e.g., Tail Recall 0.055 at $d = 5, n = 500, \beta = 3$).

- **FCI** works well in identifying skeletons and partial ancestral structures, but reading off precise arrowheads from the PAG can be more involved.

**(5) Summary of key findings.**

> In summary, ABIC LiNGAM can accurately estimate causal structures—including orientations—when errors follow a non-Gaussian distribution, even in the presence of unmeasured variables. It behaves similarly to ABIC for $\beta = 1$ (Gaussian), and approaches or slightly surpasses BANG in certain non-Gaussian conditions, especially as the dimensionality increases. Meanwhile, BANG still shows very high directional accuracy in lower-dimensional settings. Hence, our results suggest that ABIC LiNGAM is competitive with the state-of-the-art.

These observations are based on the aggregated numerical values (Figures 1–6). In particular, we highlight Arrowhead and Tail metrics because direction estimation is crucial for causal inference. By exploiting a *score-based* unified criterion rather than repeated independence tests, ABIC LiNGAM maintains stable and consistent accuracy across different sample sizes, dimensions, and non-Gaussian parameters.

Although larger samples theoretically lead to more accurate recovery of the true structure, we observed cases where increasing $n$ improves recall but lowers precision (Figure 3). Further experiments and discussions on this phenomenon are provided in Appendix.

**Key Point (10):** In many conditions, *non-Gaussian error terms* ($\beta \neq 1$) can help *ABIC LiNGAM* surpass purely Gaussian-based methods (e.g., ABIC) in *direction identification*, because they break the Markov equivalences that persist under Gaussian assumptions.

**Key Point (11):** Our experiments suggest that as *dimensionality increases*, the performance gap between *ABIC LiNGAM* and *BANG* often *narrows*, indicating *potential scalability* and robust performance of the proposed approach in higher-dimensional settings.

### 5.2 Performance Evaluation on Real-world Data

We evaluated our proposed method using a sociological data repository (https://gss.norc.org/), which has also been studied in the context of DirectLiNGAM (Shimizu et al., 2011). The dataset contains 1380 samples of sociological variables, such as parental education/occupation and offspring's outcomes. In this study, we focus on the subset of variables and their presumed causal directions as depicted in Figure 8, following the domain knowledge and temporal ordering from Duncan & Featherman (1972).

**Compared Methods.** Similar to our simulation experiments, we compared:

- **bow-free ABIC LiNGAM (proposed)**, which handles non-Gaussian errors under a bow-free ADMG constraint;

- **bow-free ABIC** (Bhattacharya et al., 2021), the baseline Gaussian version;

- **FCI** (Spirtes et al., 2000), a constraint-based algorithm for ADMGs, producing a partial ancestral graph (PAG);

- **BANG** (Wang & Drton, 2024), which also identifies bow-free ADMG directions using non-Gaussianity at a 5% significance level.

These methods were applied in the same fashion as in the simulation, focusing on how well they recover known directions and possible confounding relationships.

**Estimation of the Shape Parameter $\beta$.**  To remain consistent with our earlier assumption that the error terms follow a *multivariate generalized normal distribution (MGGD)*, we needed to estimate the shape parameter $\beta$ from the observed GSS data. Initially, we attempted to estimate a single global $\beta$ using the entire dataset, but the resulting value was below 1 and produced unstable estimates ($\delta$ became all zeros, and $\Omega$ turned into a near-diagonal matrix). Instead, we adopted a *variable-by-variable* estimation approach:

1. For each variable $X_i$, fit an MGGD-based model separately to approximate its marginal distribution and infer an individual shape parameter $\hat{\beta}_i$.

2. Choose the maximum among $\{\hat{\beta}_i\}$ as a conservative estimate, $\hat{\beta} = \max_i \hat{\beta}_i$.

3. Use this $\hat{\beta}$ as the shape parameter for the error distribution in the bow-free ADMG model.

This procedure avoided the instability observed in the global estimation method, ensuring all variables were effectively represented in the final choice of $\beta$.

**Implementation Details.**  With $\hat{\beta}$ fixed, we ran bow-free ABIC LiNGAM under the same augmented Lagrangian optimization framework described previously. For ABIC, BANG, and FCI, we applied their respective default or recommended settings, mirroring the simulation setup. Finally, we evaluated each learned graph against the domain-consistent directions in Figure 8.

**Remarks.**  By leveraging both domain knowledge (Duncan & Featherman, 1972) and our non-Gaussian ADMG framework, we expected ABIC LiNGAM to capture latent confounders via bidirected edges and to estimate orientations more accurately than purely Gaussian methods. In the next section, we present the quantitative results and discuss how they compare to the simulation findings.

Table 2: Experimental Results on Bollen Data

| Method | SKELETON | | | ARROWHEAD | | | TAIL | | |
|---|---|---|---|---|---|---|---|---|---|
| | Recall | Precision | F1-score | Recall | Precision | F1-score | Recall | Precision | F1-score |
| FCI | 0.727 | 1.000 | 0.842 | 0.643 | 0.692 | 0.666 | 0.250 | 0.667 | 0.363 |
| BANG | 0.727 | 0.889 | 0.800 | 0.643 | 0.643 | 0.643 | 0.250 | 0.500 | 0.333 |
| ABIC | 0.455 | 1.000 | 0.625 | 0.429 | 0.600 | 0.500 | 0.000 | 0.000 | 0.000 |
| ABIC LiNGAM beta est | 0.818 | 1.000 | 0.900 | 0.714 | 0.800 | 0.740 | 0.500 | 0.800 | 0.615 |

Table2 reports the performance metrics for **bow-free ABIC LiNGAM (beta est)** compared to other methods. We focus on three evaluation criteria: *Skeleton*, *Arrowhead*, and *Tail*.

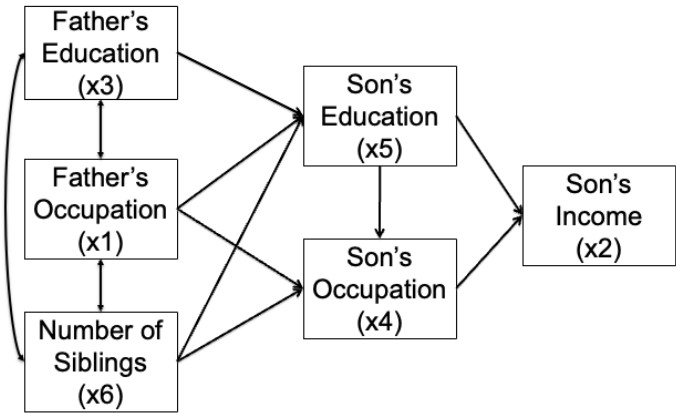

Figure 8: Variables and causal relations in the General Social Survey dataset used for the evaluation.

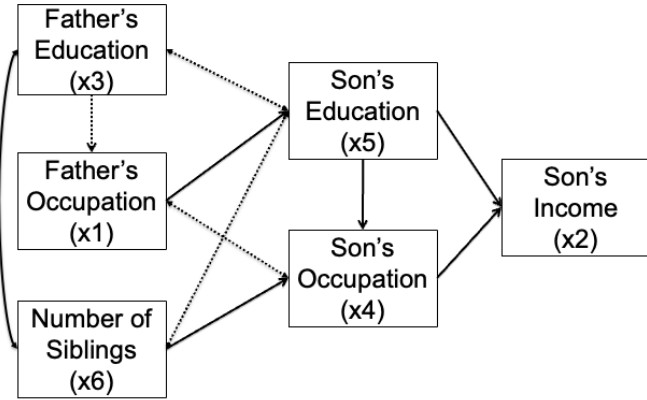

Figure 9: Causal graph produced by ABIC LiNGAM: The dashed lines represent predicted arrows that differ from the true arrows.

- **Skeleton Performance.** Our proposed method achieves a Recall of 0.818, Precision of 1.000, and an F1-score of 0.900, surpassing all baseline methods in identifying the presence or absence of edges. In contrast, ABIC attains Recall of 0.455 with an F1 of 0.625. This highlights that *ABIC LiNGAM (beta est)* can more consistently recover the correct adjacency relationships than the purely Gaussian version of ABIC.

- **Arrowhead and Tail Performance (Direction).** When examining directed-edge orientation, *ABIC LiNGAM (beta est)* also shows promising results:

  - *Arrowhead:* Recall = 0.714, Precision = 0.800, F1 = 0.740. Notably, this exceeds BANG's Arrowhead metrics (Recall 0.643, Precision 0.643, F1 0.643).
  - *Tail:* Recall = 0.500, Precision = 0.800, F1 = 0.615. Again, this is higher than BANG's Recall (0.250), Precision (0.500), and F1 (0.333).

  As such, even for orientation-specific measures, *ABIC LiNGAM (beta est)* appears to outperform both BANG and ABIC on this particular dataset. Figure **??** (formerly Figure 9) suggests that the true causal structure is largely recovered, including plausible directionality.

Overall, these results demonstrate that **ABIC LiNGAM (beta est)** can estimate causal structures (including edge directions) with higher accuracy than competing methods on the GSS dataset. In particular, its *Skeleton* F1-score (0.900) and the improvement in *Arrowhead/Tail* measures over BANG and ABIC indicate the potential effectiveness of adopting a non-Gaussian error assumption in a score-based, bow-free ADMG framework. This is consistent with our simulation findings that exploiting non-Gaussianity is crucial for precise direction inference. **ABIC LiNGAM (beta est)** can flexibly incorporate prior knowledge. In this dataset as well, incorporating prior knowledge led to improved accuracy. See Appendix for further details.

**Intuition:** The shape parameter $\beta$ can significantly *influence direction recovery*. In particular, when $\beta > 1$, the distribution exhibits *heavier tails or skewness*, thereby highlighting causal directions that might remain *indistinguishable* under purely Gaussian assumptions ($\beta = 1$).

## 6 Conclusion

In this study, we built on the score-based continuous optimization method proposed by Bhattacharya et al. (2021) and introduced **ABIC LiNGAM**, an extension of LiNGAM designed for causal structure estimation in the presence of unmeasured variables. By assuming that error terms follow a multivariate generalized normal distribution, we demonstrated that we can potentially identify not only the Markov equivalence class but also the directions in the causal structure. Moreover, because ABIC LiNGAM can accurately estimate the *SKELETON* even when the data follow a Gaussian distribution, our approach can be viewed as a more generalized version of the method in Bhattacharya et al. (2021).

We also proved that parameters in bow-free ADMGs are almost everywhere identifiable from the covariance matrix under the multivariate generalized normal distribution, thereby extending the results of Brito & Pearl (2002) to non-Gaussian settings. Through simulations and experiments with real-world data, we confirmed that the proposed method achieves accuracy comparable to existing methods (such as BANG or FCI) in recovering causal structures, including causal directions. These

findings suggest that our approach provides a useful framework for causal discovery in practical settings where unmeasured variables may be present.

Concretely, our main contributions can be summarized as follows:

- **Direction Identification under Non-Gaussianity:** Whereas many Gaussian-based methods can only recover the Markov equivalence class, our method leverages the multivariate generalized normal distribution ($\beta \neq 1$) to rigorously identify causal directions.

- **Theoretical Identifiability:** We proved that parameters in bow-free ADMGs are almost everywhere identifiable from the covariance matrix, thereby extending the results of Brito & Pearl (2002) to non-Gaussian settings.

- **Empirical Validation:** Using both simulations and real datasets, we showed that our method achieves accuracy **comparable** to that of existing methods (such as BANG or FCI) in recovering causal structures and directions.

In future work, we will explore ways to **reduce estimation time**, investigate **extensions to nonlinear data structures**, and apply our method to **mixed data** that include discrete variables. Advancing these directions will help establish a more efficient and versatile methodology for causal discovery across diverse real-world scenarios.

In this paper, we extended the differentiable score-based approach from the Gaussian to the non-Gaussian setting. This does not imply removing all assumptions; rather, the non-Gaussian error terms furnish additional identifiable information beyond second moments, thereby enabling us to determine causal directions more strictly. The results underscore that adopting non-Gaussianity is not just a weaker constraint but a critical tool to enhance identifiability in causal discovery under latent confounding.

## 7 Acknowledgments

This work was supported by the Japan Science and Technology Agency (JST) under CREST Grant Number JPMJCR22D2.

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

## A    Appendix

### A.1    Proof of the Identifiability of Parameters in Bow-Free ADMGs with Multivariate Generalized Normal Distributions

**Theorem 1**

Let $G$ be a bow-free ADMGs with error terms following a multivariate generalized normal distribution, and let the set of parameters of $G$ be $\theta = \{\delta, \Omega\}$. Then, for almost all $\theta$, the following holds:

$$\Sigma(\theta) = \Sigma(\theta')$$

implies $\theta = \theta'$.

In other words, if two parameter sets $\theta$ and $\theta'$ give the same covariance matrix $\Sigma$, then $\theta$ and $\theta'$ must be identical, except possibly when $\theta$ belongs to a set of Lebesgue measure zero. If the two lemmas described later can be proven for the case where the error terms follow a multivariate generalized normal distribution, this theorem can be demonstrated using the same proof as in Brito & Pearl (2002).

**Definition 1**

A *path* in a graph is a sequence of edges (directed or bidirectional), where each edge starts from the node where the previous edge ends. A *directed path* consists only of directed edges all pointing in the same direction. A node $X$ is called an *ancestor* of a node $Y$ if there is a directed path from $X$ to $Y$. A path is said to be *blocked* if there is a node $Z$ on the path such that there are consecutive edges pointing toward $Z$ (e.g., $\cdots \to Z \leftarrow \dots$). In this case, $Z$ is called a *collider*.

**Definition 2**

In a DAG, the *depth* of a node is defined as the length (number of edges) of the longest path directed from its ancestors to that node.

**Lemma 1**

Let $X$ and $Y$ be the nodes in a bow-free ADMG with $depth(X) \geq depth(Y)$. Then, all paths between $X$ and $Y$ that include a node $Z$ satisfying $depth(Z) \geq depth(X)$ are blocked by colliders. This lemma is based on graph theory and does not depend on the distribution of the error terms. It is quoted from Brito & Pearl (2002).

**Definition 3**

For each node $Y$, the set of edges directed to $Y$, denoted by $I(Y)$, is defined as the union of the following two sets: (a) the set of all directed edges pointing to $Y$, (b) the set of all bidirectional edges between $X$ and $Y$, where $depth(X) < depth(Y)$.

**Lemma 2**

Let $Y$ be a variable at depth $k$ in a bow-free ADMG. Assume that the parameters of all edges connecting variables of a depth less than $k$ are identifiable. Then, in almost all cases, the parameters of each edge in the set $I(Y)$ are identifiable.

*Proof.* In Brito & Pearl (2002), the identifiability of bow-free models was established under the assumption that the error terms follow a multivariate normal distribution. In this study, we extend this identifiability result to the case where the error terms follow the aforementioned multivariate generalized normal distribution. The proof itself draws heavily from Brito & Pearl (2002).

Wright's method Wright (1960) relies on linear relationships between variables and covariance structures. Since the multivariate generalized normal distribution is closed under linear transformations((Gómez et al., 1998)), Wright's method is applicable beyond the normal distribution as long as the necessary linear conditions are satisfied. Indeed, Wright (1960) also mentions that Wright's method can be applied to distributions other than the normal distribution.

Let $X = \{X_1, X_2, \ldots, X_m\}$ be the set of variables with a depth less than $k$, and suppose that these variables are connected to $Y$ by directed or undirected edges. By the properties of bow-free ADMGs, a one-to-one correspondence exists between each variable in $X$ and the edges in $I(Y)$. Therefore, $I(Y)$ can be expressed as

$$I(Y) = \{(X_1, Y), (X_2, Y), \ldots, (X_m, Y)\}.$$

Applying Wright's method to each pair $(X_i, Y)$ yields the following equations:

$$\sigma_{X_iY} = \sum_{p_i} T(p_i), \quad i = 1, \ldots, m$$

where $\sigma_{X_iY}$ represents the covariance between $X_i$ and $Y$, the sum is over all paths $p_i$ between $X_i$ and $Y$ that have direct or indirect effects or associations, and $T(p_i)$ represents the product of parameters along the path $p_i$.

For each $i$, let $\lambda_i$ be the parameter corresponding to the edge $(X_i, Y)$. The equation can be rewritten as

$$\sigma_{X_iY} = \lambda_i + \sum_{j \neq i} \lambda_j a_{ij}, \quad i = 1, \ldots, m$$

where the coefficients $a_{ij}$ are functions of identifiable parameters corresponding to edges connecting variables of a depth less than $k$. These coefficients reflect contributions from direct or indirect effects or associations involving known parameters, excluding the direct edge $(X_i, Y)$.

Under the assumption, by the induction hypothesis, that all parameters of edges connecting variables of a depth less than $k$ are identifiable, the coefficients $a_{ij}$ are known quantities. Therefore, we obtain a system of $m$ linear equations with $m$ unknowns $\lambda_1, \ldots, \lambda_m$, which can be written in a matrix form as

$$\sigma = A\lambda \tag{24}$$

where

$$\sigma = \begin{pmatrix} \sigma_{X_1 Y} \\ \sigma_{X_2 Y} \\ \vdots \\ \sigma_{X_m Y} \end{pmatrix}, \quad \lambda = \begin{pmatrix} \lambda_1 \\ \lambda_2 \\ \vdots \\ \lambda_m \end{pmatrix}, \quad A = \begin{pmatrix} 1 & a_{12} & \ldots & a_{1m} \\ a_{21} & 1 & \ldots & a_{2m} \\ \vdots & \vdots & \ddots & \vdots \\ a_{m1} & a_{m2} & \ldots & 1 \end{pmatrix}.$$

To establish the identifiability of the parameters $\lambda_i$, it suffices to show that matrix $A$ is invertible in almost all cases, that is, $\det(A) \neq 0$ except on a set of measure zero, considering that the left-hand side $\sigma$ is observable. The matrix $A$ has all diagonal elements equal to 1, and off-diagonal elements depending on the model parameters. The determinant can be expressed in terms of the diagonal and off-diagonal elements, as shown in (25).

$$\det(A) = 1 + T, \tag{25}$$

where $T$ is either zero or a polynomial in the model parameters that do not contain any constant term.

According to a well-known result in algebraic geometry Okamoto (1973), the set of parameter values where $\det(A) = 0$ has Lebesgue measure zero in the parameter space. This is because $\det(A) = 0$ defines an algebraic variety of a lower dimension within the parameter space. Therefore, the matrix $A$ is invertible in almost all cases, and the system of linear equations has a unique solution.

Thus, under the given assumptions, each parameter $\lambda_i$ is identifiable in almost all cases.

$\square$

## A.2 Hölder's Inequality

Hölder's inequality is a fundamental result in analysis that provides estimates for sequences (or more generally, measurable functions on a measure space $(\Omega, \mu)$) in terms of their $L^p$-norms. Specifically, for $p, q \geq 1$ satisfying $\frac{1}{p} + \frac{1}{q} = 1$, Hölder's inequality states that for any two sequences $(a_k)$ and $(b_k)$:

$$\sum_{k=1}^{\infty} |a_k b_k| \leq \left( \sum_{k=1}^{\infty} |a_k|^p \right)^{1/p} \left( \sum_{k=1}^{\infty} |b_k|^q \right)^{1/q}. \tag{26}$$

Furthermore, by taking $b_k = 1$, we obtain a useful inequality for finite sums as follows:

$$\left( \sum_{k=1}^{n} |a_k| \right)^p \leq n^{p-1} \sum_{k=1}^{n} |a_k|^p. \tag{27}$$

This special case reflects how the $L^p$-norm behaves in a finite setting and is central to understanding the interplay between norms and summation.

In our specific problem, we use Hölder's inequality to handle the terms involving $\beta$th powers of squared residuals. Considering the log-likelihood expression after rearrangement,

$$\ell(\mu, \Sigma, \beta | X)$$

$$= -\frac{N}{2} \log \Omega_{ii.-i} - \frac{1}{2\Omega_{ii.-i}^{\beta}} \sum_{l=1}^{N} \left( \left( X_i^{(l)} - \sum_{j \in \mathrm{pa}(i)} \delta_{i,j} X_j^{(l)} - \sum_{k \in sp(i)} \Omega_{i,k} Z_k^{(l)} \right)^2 \right)^{\beta}$$

$$= -\frac{N}{2} \log \Omega_{ii.-i} - \frac{1}{2\Omega_{ii.-i}^{\beta}} \frac{N^{\beta-1}}{N^{\beta-1}} \sum_{l=1}^{N} \left( \left( X_i^{(l)} - \sum_{j \in \mathrm{pa}(i)} \delta_{i,j} X_j^{(l)} - \sum_{k \in sp(i)} \Omega_{i,k} Z_k^{(l)} \right)^2 \right)^{\beta}. \tag{28}$$

We introduce the factor $\frac{N^{\beta-1}}{N^{\beta-1}}$ to rewrite the sum in a form amenable to Hölder's inequality. Define the sequence as

$$a_l = \left| \left( X_i^{(l)} - \sum_{j \in \mathrm{pa}(i)} \delta_{i,j} X_j^{(l)} - \sum_{k \in sp(i)} \Omega_{i,k} Z_k^{(l)} \right)^2 \right|^{\beta},$$

and let $b_l = 1$. If we choose $p = \beta$, and hence $q = \frac{\beta}{\beta-1}$ (so that $\frac{1}{p} + \frac{1}{q} = 1$), Hölder's inequality gives us

$$\sum_{l=1}^{N} |a_l b_l| \leq \left( \sum_{l=1}^{N} |a_l|^{\frac{\beta}{\beta}} \right)^{\frac{1}{\beta}} \left( \sum_{l=1}^{N} |b_l|^{\frac{\beta}{\beta-1}} \right)^{\frac{\beta-1}{\beta}} = \left( \sum_{l=1}^{N} a_l \right)^{\frac{1}{\beta}} N^{\frac{\beta-1}{\beta}}.$$

Rearranging this inequality, we obtain a lower bound on $\sum_{l=1}^{N} a_l$ in terms of $N^{\beta-1}$ and the $L^\beta$-norm of the residuals

$$\sum_{l=1}^{N} \left( \left( X_i^{(l)} - \cdots \right)^2 \right)^{\beta} \geq \frac{\left( \sum_{l=1}^{N} |X_i^{(l)} - \cdots| \right)^{\beta}}{N^{\beta-1}}.$$

Substituting this bound back into the expression for $\ell(\mu, \Sigma, \beta | X)$, we have

$$\ell(\mu, \Sigma, \beta|X) \geq -\frac{N}{2}\log\Omega_{ii.-i} - \frac{1}{2\Omega_{ii.-i}^{\beta}}\frac{1}{N^{\beta-1}}\left\|X_i - \sum_{j\in\mathrm{pa}(i)}\delta_{i,j}X_j - \sum_{k\in sp(i)}\Omega_{i,k}Z_k\right\|^{2\beta}. \tag{29}$$

Thereby, Hölder's inequality is employed to provide a nontrivial lower bound on the log-likelihood by relating sums of $\beta$th powers of squared terms to the $\beta$th power of their $L^1$-norm, scaled appropriately by $N^{\beta-1}$. This facilitates a more tractable analysis of the growth behavior and bounding properties of the likelihood function.

### A.3  Additional Experiments with Larger Sample Sizes

In addition to the simulations described in Section 5 of the main text, we conducted further experiments to investigate how our score-based continuous optimization approach (*ABIC LiNGAM*) behaves under larger sample sizes. Specifically, we generated datasets of size $n \in \{100, 500, 1000, 2000, 5000\}$ with five observed variables (dimension $k = 5$) and a fixed non-Gaussian parameter $\beta = 5$. We considered both cases: beta ture (i.e., $\beta$ is known) and beta est (i.e., $\beta$ is estimated from the data). Our primary goal was to examine whether increasing $n$ leads to higher recall while potentially decreasing precision due to "over-detection" of minor dependencies.

Table 3 presents the results for three types of edges (*SKELETON, ARROWHEAD, TAIL*), evaluated by recall, precision, and F1-score. One would expect, from a theoretical standpoint, that as the sample size grows, the estimated causal structure should approach the true structure. However, as seen in Table 3, although recall does tend to increase with larger $n$, precision can decline in some settings, leading to F1-scores that do not strictly increase. This phenomenon arises because a larger sample size increases the statistical power of our score-based method, causing it to detect even subtle or spurious edges as false positives. Such a trend is common in flexible models when the penalty strength or thresholds are not tuned for very large sample sizes.

As Table 3 shows, **recall** generally improves with $n$ (i.e., fewer missed edges), but **precision** ($\approx$ the proportion of correctly identified edges among those detected) can decrease, indicating a rise in false positives. Consequently, the F1-scores (F1 $= 2 \times$ recall $\times$ precision$/($recall $+$ precision$)$) do not necessarily increase monotonically. Moreover, because our causal graph is randomly generated for each run, the "difficulty" of each dataset can vary, causing some fluctuations (e.g., at $n = 500$ vs. $n = 1000$).

These results help explain why large-sample outcomes can sometimes appear to deviate from the theoretical expectation of "convergence to the true graph." In principle, if the regularization or threshold hyperparameters are adjusted for the sample size, false positives may be reduced, thus improving precision alongside recall. However, in our current implementation, these hyperparameters are held constant, leading to potential "over-detection" when $n$ grows large.

**Relation to Figure 3.**   In Section 5 (Figure 3), the last subgraph similarly shows a case in which increasing sample size does not strictly improve F1-scores, despite recall gains. As clarified above, our score-based approach tends to pick up spurious edges with large $n$, thus reducing precision and causing a mismatch from purely theoretical expectations. We believe that this phenomenon arises from not tuning the penalty term to suppress minor spurious connections in very large samples.

Table 3: Extended experiment results of ABIC LiNGAM ($\beta = 5$) with sample sizes $n \in \{100, 500, 1000, 2000, 5000\}$ and dimension $k = 5$. Metrics are reported for three edge categories (*SKELETON, ARROWHEAD, TAIL*), under both the "$\beta$-est" (top) and "$\beta$-true" (bottom) conditions. Numbers are averaged over multiple random simulations. Note that data generation relies on randomly assigned causal graphs and random noise, causing minor variability across runs.

| Sample Size | SKELETON | | | ARROWHEAD | | | TAIL | | |
|---|---|---|---|---|---|---|---|---|---|
| | Recall | Precision | F1 | Recall | Precision | F1 | Recall | Precision | F1 |
| *(a) ABIC LiNGAM beta true* | | | | | | | | | |
| $n = 100$ | 0.903 | 0.776 | 0.826 | 0.722 | 0.682 | 0.691 | 0.669 | 0.443 | 0.502 |
| $n = 500$ | 0.948 | 0.774 | 0.843 | 0.727 | 0.676 | 0.691 | 0.643 | 0.357 | 0.430 |
| $n = 1000$ | 0.965 | 0.695 | 0.800 | 0.739 | 0.584 | 0.641 | 0.647 | 0.315 | 0.404 |
| $n = 2000$ | 0.985 | 0.713 | 0.821 | 0.768 | 0.619 | 0.673 | 0.699 | 0.328 | 0.429 |
| $n = 5000$ | 0.994 | 0.720 | 0.826 | 0.771 | 0.597 | 0.661 | 0.673 | 0.303 | 0.403 |
| *(b) ABIC LiNGAM beta est* | | | | | | | | | |
| $n = 100$ | 0.883 | 0.806 | 0.832 | 0.706 | 0.682 | 0.680 | 0.613 | 0.452 | 0.490 |
| $n = 500$ | 0.942 | 0.821 | 0.869 | 0.742 | 0.727 | 0.720 | 0.673 | 0.409 | 0.485 |
| $n = 1000$ | 0.928 | 0.727 | 0.808 | 0.739 | 0.634 | 0.672 | 0.603 | 0.359 | 0.431 |
| $n = 2000$ | 0.953 | 0.750 | 0.833 | 0.734 | 0.670 | 0.689 | 0.660 | 0.364 | 0.451 |
| $n = 5000$ | 0.992 | 0.731 | 0.832 | 0.744 | 0.616 | 0.660 | 0.711 | 0.324 | 0.424 |

Future work will focus on dynamically adjusting thresholds and penalties to maintain a balance between recall and precision, ensuring more consistent alignment with the theoretical asymptotic behavior.

## A.4 Performance Evaluation on Real Data Using Prior Knowledge

In ABIC LiNGAM, prior knowledge can be incorporated into the inference process following the code implementation of ABIC available at https://gitlab.com/rbhatta8/dcd. Specifically, by defining hierarchical causal orders (tiers) and incorporating prior knowledge that certain variables are unconfounded, we can restrict the parameter ranges (bounds) considered during the estimation. Consequently, edges that contradict the causal order, as well as bidirectional edges among unconfounded variables, are represented with predetermined constraints in the parameter space (e.g., fixed to zero) and are thus automatically excluded during the inference process. Thereby, explicitly reflecting prior knowledge in the model ensures that assumptions regarding causal directionality and the absence of confounding factors are maintained, enabling an efficient structural estimation.

For example, the General Social Survey dataset obtained from a sociological data repository (https://gss.norc.org/) includes variables pertaining to parents and children. If a causal path exists from the parents to the children, a causal path from the children to the parents is not possible. In the following section, we consider the inferences with a hierarchical causal structure. Specifically, we divided the variables into two groups with a two-tiered structure—Father's Occupation, Father's Education and Number of Siblings, Son's Education, Son's Occupation, Son's Income—and prohibited the existence of directed edges from children's variables to parents' variables.

As demonstrated by the results shown in Figure 10 and Table 4, incorporating prior information into ABIC LiNGAM leads to a higher estimation accuracy compared to the approach without such

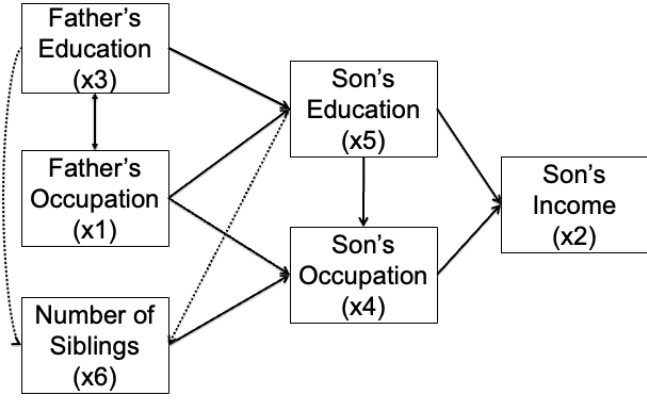

Figure 10: Causal graph produced by ABIC LiNGAM incorporating prior knowledge: The dashed lines represent predicted arrows that differ from the true arrows.

Table 4: Experimental Results for Bollen Data incorporating prior knowledge

| Method | SKELETON | | | ARROWHEAD | | | TAIL | | |
|---|---|---|---|---|---|---|---|---|---|
| | Recall | Precision | F1-score | Recall | Precision | F1-score | Recall | Precision | F1-score |
| ABIC LiNGAM beta est | 0.818 | 1.000 | 0.900 | 0.714 | 0.800 | 0.740 | 0.500 | 0.800 | 0.615 |
| ABIC LiNGAM beta est prior knowledge | 0.818 | 1.000 | 0.900 | 0.692 | 0.900 | 0.782 | 0.857 | 0.750 | 0.800 |

information. The estimated graph aligns closely with the presumed true causal structure. This suggests that incorporating prior knowledge into ABIC LiNGAM is feasible and that leveraging such information can yield improved accuracy in practical applications.

