# OpenReview forum: "Differentiable Causal Discovery of Linear Non-Gaussian Acyclic Models Under Unmeasured Confounding"
_TMLR — Accepted by TMLR_

### Review · Reviewer_4ipP · 2025-03-12

**Summary Of Contributions:**

In this paper, the authors present ABIC LINGAM, a novel method that brings insights from recent continuous optimization-based approaches to graph structure learning to the non-Gaussian setting, extending to multivariate generalized normally distributed error terms. In doing so, the proposed methods go beyond just Markov equivalence classes, breaking what would otherwise be symetric in the likelihood under Gaussian errors. The authors prove identification, and present experiments validating their approach.

**Audience:**

Yes

**Broader Impact Concerns:**

I have no broader impacts concerns.

**Claims And Evidence:**

Yes

**Requested Changes:**

See Weaknesses above.

I don't see a section in this review portal for clarifying questions, so I will ask here.

Could the authors please comment on how ABIC LINGAM will perform when the parametric assumptions of multivariate generalized normally distributed error terms are violated? Specifically, what happens when errors are actually Gaussian? What about when structural functions are nonlinear? What about structural functions are linear, but errors are non-Gaussian in some other way?

**Strengths And Weaknesses:**

Strengths:

The paper is overall strong, although I will state upfront that causal discovery is slightly outside my primary area of expertise. I will let other reviewers comment on the technical details of proofs. That said, the authors make tightly scoped technical claims, provide what appears to be sufficient theoretical and empirical evidence, and provide a thorough discussion of how this work fits within the broader context of structure learning. From a somewhat outsiders perspective, it appears to be a natural extension of two existing and powerful ideas; casting structure learning as continuous optimization and exploiting non-Gaussian noise to disambiguate edge orientation.

Weaknesses:
While the paper is presented clearly, it's not clear how the empirical study substantiates the claims made. The differences between ABIC LINGAM and BANG appears very minimal in almost all of the empirical results. Is there another reason one should prefer ABIC LINGAM? The results do appear to be better in the "real data".


At various points it was a bit difficult to distinguish between what was background and what were novel contributions. Concretely, I would recommend moving some of the discussion of Bhattacharya et al. (2021) in Section 4.2 to an early section on Background, but this is really a minor suggestion.

---

> ### Author Response · Authors · 2025-03-20
> **Response to the comments received part1**
>
> Dear Reviewer,
>
> Thank you very much for taking the time to read our manuscript and provide valuable feedback. We appreciate your thoughtful comments and suggestions, which have helped us clarify and strengthen the presentation of our work. Below, we address your main points in detail.
>
> Due to the character limit, I'll split my response into two parts.
>
> ①「Weaknesses: While the paper is presented clearly, it's not clear how the empirical study substantiates the claims made. The differences between ABIC LINGAM and BANG appears very minimal in almost all of the empirical results. Is there another reason one should prefer ABIC LINGAM? The results do appear to be better in the "real data".」
>
> Although the performance differences are small, the continuous optimization (score-based) approach ABIC LiNGAM offers the following benefits:
>
> (A) Consistent Optimization Framework
> BANG is a constraint-based approach that statistically tests constraints (e.g., conditional independence) and identifies a graph structure that satisfies them. This can lead to issues such as Type I/Type II errors at each testing stage and the need for multiple tests. In contrast, ABIC LiNGAM is formulated by maximizing a single score (e.g., log-likelihood) at once, treating all parameters in one optimization problem. This unified approach simplifies both implementation and operation.
>
> (B) Flexible Incorporation of Prior Knowledge
> By imposing numerical constraints, it is straightforward to incorporate specific domain knowledge (e.g., hierarchical structures or edges to exclude), facilitating extensions to more complex data.
>
> (C) Unified Handling of Gaussian and Non-Gaussian Distributions
> Assuming a generalized normal distribution for the error terms allows for both Gaussian and non-Gaussian cases to be handled within the same framework.
> - For Gaussian distributions: estimation is limited to the Markov equivalence class (similar to existing ABIC).
> - For non-Gaussian distributions: edge directions can also be determined.
>
> (D) Extensibility
> Incorporating nonlinear structures or mixed-type data in constraint-based approaches often requires redesigning statistical tests. Score-based approaches, however, can easily accommodate extensions by modifying the loss function or regularization terms.
>
> For these reasons, even if simulation accuracy is comparable to that of existing methods, we consider the continuous optimization approach and its potential for future extensions to be a key advantage of our proposed method.
>
> ②「At various points it was a bit difficult to distinguish between what was background and what were novel contributions. Concretely, I would recommend moving some of the discussion of Bhattacharya et al. (2021) in Section 4.2 to an early section on Background, but this is really a minor suggestion.」
>
> In the current version of the paper, details of the ABIC (Augmented BIC) algorithm proposed by Bhattacharya et al. (2021) were presented alongside the core of our proposed method in the same section. This made it difficult to distinguish our novel contributions. Therefore, we are planning the following reorganization:
>
> (A) Consolidating the algorithm overview in Section 2 (Previous Research)
> We will move the ABIC overview to Section 2, where we discuss existing methods, to provide a concise summary of what ABIC entails.
>
> (B)  Focusing on essential points in Section 4.2 (Extensions of the Proposed Method)
> To highlight the new elements of ABIC LiNGAM, Section 4.2 will focus solely on extensions we propose—such as handling non-Gaussian errors and generalized normal distribution parameters. Since the algorithmic details from Bhattacharya et al. (2021) are their own contribution, we will avoid duplicating that content in our paper.
>
> ③「Could the authors please comment on how ABIC LINGAM will perform when the parametric assumptions of multivariate generalized normally distributed error terms are violated? Specifically, what happens when errors are actually Gaussian?  」
>
> Conclusion:
> When the error terms follow a Gaussian distribution, ABIC LiNGAM (like other Gaussian linear methods, e.g., Bhattacharya et al. (2021)) can only identify one member of the Markov equivalence class. As a result, individual causal directions are not uniquely determined. However, our approach unifies Gaussian and non-Gaussian (generalized normal) error distributions in a single framework. Therefore, even if the true data-generating process is Gaussian, the same framework can still be applied consistently.

---

> > ### Author Response · Authors · 2025-03-20
> > **Response to the comments received part2**
> >
> > Dear Reviewer,
> >
> > ④「What about when structural functions are nonlinear?」
> >
> > Conclusion: Although this study is based on a theoretical framework assuming linear models, additional simulations confirmed that even for nonlinear structural functions such as polynomial terms (for example, Yi = sum(delta * Xj^2) + epsilon_i), it is possible to estimate the skeleton (which variables are connected) with a certain degree of accuracy. On the other hand, for estimating the direction of edges (TAIL), there were cases where the existing method (BANG) achieved higher accuracy, indicating that further research is necessary to theoretically guarantee performance on general nonlinear models.
> >
> > Details:
> > (A)  Nonlinear Scenario and Simulation Setup: We assumed a polynomial model including squared terms, for instance, Yi = sum(delta * Xj^2) + epsilon_i, even though our theory fundamentally assumes linear models. We then conducted additional simulations to verify how well the proposed method (ABIC LiNGAM) could estimate the structure in such a scenario.
> > (B)  Experimental Results: For example, with five variables and a sample size of 500, the skeleton F-value was highest for ABIC LiNGAM (0.901) when the shape parameter beta was given the true value, demonstrating that this method can effectively estimate the presence or absence of connections even under nonlinearity. However, for TAIL (the direction of edges), BANG (0.480) outperformed our method (0.035 to 0.045), showing that BANG can yield superior edge-direction estimates in some nonlinear scenarios.
> > (C) Discussion (Challenges Beyond Linear Theory):
> > Our theoretical framework assumes “linear structure + non-Gaussian errors,” so it does not rigorously guarantee identifiability or convergence for general nonlinear models. Nevertheless, our additional simulations suggest that the skeleton can be adequately recovered. Improving edge-direction accuracy for nonlinear causal structures remains a challenge, and we plan to address this by exploring nonlinear extensions to the loss function and regularization, as well as theoretical guarantees for identification under more general assumptions.
> >
> > ⑤「 What about structural functions are linear, but errors are non-Gaussian in some other way?」
> >
> > Conclusion:
> > Even when the structural function is linear but errors follow a non-Gaussian distribution (e.g., a multivariate t distribution), simulations show that ABIC LiNGAM can estimate the causal structure reasonably well. However, if the shape parameter (beta) is inaccurately estimated, the overall accuracy of causal structure inference can decline, indicating the need for caution in practice.
> >
> > Details:
> > (A)  Experiment Setup:
> > We employed a multivariate t distribution with 3 degrees of freedom for the error terms in a five-variable system at a sample size of 500. We compared two scenarios: (a) beta fixed at 3, and (b) beta estimated from the data.
> >
> > (B)  Results:
> > When beta = 3 was fixed, ABIC LiNGAM provided the best results for skeleton, arrow, and tail accuracy. However, in the scenario where beta was estimated, its inferred value sometimes deviated from the true value, leading to a notable drop in the accuracy of causal direction estimates.
> >
> > (C)   Practical Considerations:
> > Accurate estimation of the shape parameter is crucial when assuming heavy-tailed distributions. In practice, testing multiple candidate values or comparing models with criteria such as AIC or BIC may improve robustness. If prior domain knowledge about the approximate degrees of freedom is available, supplying that information to ABIC LiNGAM can enhance the reliability of causal direction estimation.

---

### Review · Reviewer_DUVv · 2025-03-21

**Summary Of Contributions:**

The authors propose a novel score-based method, ABIC-LiNGAM, for identifying causal structures with latent variables under the linear non-Gaussian acyclic model (LiNGAM). By assuming that the error terms follow a multivariate generalized normal distribution, the method enables the estimation of not only the Markov equivalence class, but also the causal directions within the structure. The authors further provide theoretical guarantees by proving the identifiability of parameters in acyclic directed mixed graphs (ADMGs) under the assumed error distribution. To demonstrate the effectiveness of their approach, they conduct extensive simulation studies as well as experiments on real-world datasets.

**Audience:**

Yes

**Claims And Evidence:**

Yes

**Requested Changes:**

1. The manuscript includes a limited discussion of relevant literature. It would benefit from citing more related work on score-based methods for causal discovery, particularly earlier contributions on identifying ADMGs. For the cited works discussed in more detail (e.g., Wang & Drton (2024); Bhattacharya et al. (2021)), the authors should explicitly clarify the differences from the present study and highlight the contributions and advantages of their method.

2. There are several instances of repetitive descriptions that could be streamlined. For example, the statement “Brito & Pearl (2002) demonstrated that a bow-free ADMG model is almost always identifiable from the observed covariance matrix” appears in both Section 2.3 and Section 2.31. In addition, some key technical terms lack sufficient explanation and should be better clarified for accessibility.

3. Some elements, such as Algorithm 2 (ABIC-LiNGAM), appear to be reused or closely resemble content from the article Differentiable Causal Discovery Under Unmeasured Confounding. The authors should clarify whether these are original contributions or adapted from prior work.

4. In Equation (15), is the notation $Z_{-i}$ consistent with the $Z_{-i}$ used in the first line following the equation? Clarification would help prevent potential confusion.

5. The trace operator notation, $tr()$, should be clearly defined when first introduced, such as in Equation (22).

6. The algorithms presented would benefit from more detailed explanations or better formatting (e.g., spacing or step annotations) to improve readability. Specifically, the differences between Algorithm 1 and Algorithm 2 should be clearly articulated.

7. Some descriptions in the manuscript lack structure. It is recommended to use "Remarks" to emphasize key references, results, or insights to enhance the clarity and readability of the manuscript.

8. The simulation results do not appear to exhibit expected large-sample properties. For instance, the last subgraph in Figure 3 seems inconsistent with theoretical expectations. Can the authors provide further explanation?

9. It would strengthen the theoretical contribution if the authors could include an additional theorem to show that ABIC-LiNGAM can consistently recover the skeleton of the causal graph.

**Strengths And Weaknesses:**

**Strengths**

1. The manuscript addresses the important and challenging problem of causal structure estimation in the presence of unmeasured confounders.

2. The authors propose a score-based method for estimating bow-free acyclic directed mixed graphs (ADMGs) under a linear non-Gaussian acyclic model.

3. They also provide theoretical guarantees by proving identifiability results when the error terms follow a multivariate generalized normal distribution.

4. The overall structure of the manuscript is clear and well-organized.

**Weaknesses**

A notable limitation of the study is the simplicity of the model setup. In particular, the assumption that the noise terms follow a multivariate generalized normal distribution may be too restrictive and could limit the generalizability of the proposed method.

---

> ### Author Response · Authors · 2025-04-13
> **Response to the comments received part1**
>
> Dear Reviewers,
>
> Thank you very much for taking the time to thoroughly read our manuscript and for providing valuable feedback. Your comments and suggestions have been extremely helpful in clarifying and strengthening our presentation of the research. In the following, we outline our responses and revisions regarding the nine points you raised.
>
> ①「The manuscript includes a limited discussion of relevant literature. It would benefit from citing more related work on score-based methods for causal discovery, particularly earlier contributions on identifying ADMGs. For the cited works discussed in more detail (e.g., Wang & Drton (2024); Bhattacharya et al. (2021)), the authors should explicitly clarify the differences from the present study and highlight the contributions and advantages of their method.」
>
> Below are the main points of revision we made, aiming to reinforce the discussion of related work and clarify the positioning of our study.
>
> 1.Addition of References and Expanded Discussion
> We have newly cited prior studies on the identification of ADMGs (e.g., Richardson & Spirtes, 2002; Tashiro et al., 2014) as well as literature on score-based methods (e.g., Nowzohour et al., 2017; Bernstein et al., 2020; Chen et al., 2021; Claassen & Bucur, 2022; Ng et al., 2024), and supplemented our comparison and discussion of each approach.
>
> 2.Comparison with Wang & Drton (2024) and Bhattacharya et al. (2021)
> We clarified the differences between the methods proposed in Wang & Drton (2024) (BANG) and Bhattacharya et al. (2021) (ABIC), and explained in detail how our research complements or extends these previous works. In particular, we emphasized that our method leverages non-Gaussianity to achieve precise identification of causal directions.
>
> 3.Emphasis on the Contributions and Advantages of This Study
> Based on the additional literature review and comparative findings, we reiterated that our work is the first framework within score-based methods assuming non-Gaussianity that fully identifies Bow-free ADMGs. We have also presented the theoretical underpinnings (proof of identifiability) and experimental validations (applicability to large-scale data).
>
> With these revisions, we believe that the relationships between our study and existing research, as well as the novelty and effectiveness of our approach, have been more clearly articulated.
>
> ②「There are several instances of repetitive descriptions that could be streamlined. For example, the statement “Brito & Pearl (2002) demonstrated that a bow-free ADMG model is almost always identifiable from the observed covariance matrix” appears in both Section 2.3 and Section 2.31. In addition, some key technical terms lack sufficient explanation and should be better clarified for accessibility.」
>
> Below are the revisions we made to address the noted issues:
>
> 1.Elimination of Redundancy
> As pointed out, the discussion on identifiability by Brito & Pearl (2002) was repeated in both Section 2.3 and Section 2.31. To resolve this, we now provide only a high-level overview in the higher-level section (Section 2.3), while consolidating the detailed arguments, proofs, and extensions of Brito & Pearl (2002) in the lower-level section (Section 2.31). In Section 2.3, we simply state that “bow-free ADMG models are almost always identifiable” and then refer the reader to the subsequent section for more details.
>
> 2.Additional Terminology Explanations
> To ensure clarity for the reader regarding bow-free ADMG, Markov equivalence class, and Non-Gaussianity, we have added supplementary explanations. Specifically, we created a subsection titled “Definition and Key Terms,” which succinctly defines each concept and provides relevant background.
>
> 3.Conciseness and Reorganization
> We have consolidated the discussion on extending from Gaussian to MGGD (multivariate generalized normal distribution) into a single paragraph, so that it clearly illustrates how the scope of discussion expands from the traditional Gaussian assumption.
>
> In addition, we have explicitly highlighted the importance of exploiting non-Gaussianity for causal direction estimation, noting briefly that our approach can estimate structures that would be indistinguishable under previous methods.

---

> > ### Author Response · Authors · 2025-04-13
> > **Response to the comments received part2**
> >
> > ③「Some elements, such as Algorithm 2 (ABIC-LiNGAM), appear to be reused or closely resemble content from the article Differentiable Causal Discovery Under Unmeasured Confounding. The authors should clarify whether these are original contributions or adapted from prior work.」
> >
> > "Below is our plan to reorganize the content to clarify “what constitutes the new contribution” of our study, addressing the overlap between the explanation of the ABIC (Augmented BIC) algorithm by Bhattacharya et al. (2021) as prior work and the core part of our proposed method within the same section.
> >
> > Section 1.2 (Related Work) will consolidate the algorithm overview
> > We intend to include a concise description of ABIC as part of the “background on existing methods” in Section 1.2. This way, readers can easily understand what ABIC is and how it operates without conflating it with our novel contributions.
> >
> > Section 4.2 (Extensions in Our Proposed Method) will focus on essential new elements only
> > Since our proposal is an extension called “ABIC LiNGAM,” Section 4.2 will highlight only the newly added features (e.g., handling non-Gaussian errors, managing parameters from the generalized normal distribution). The detailed mechanics of Bhattacharya et al. (2021)’s algorithm itself—which belongs to prior work—will not be repeated in this manuscript. Instead, we will build on their contributions and focus on explaining our enhancements."
> >
> > ④「In Equation (15), is the notation Z_{-i}
> >  consistent with the Z_{-i}
> >  used in the first line following the equation? Clarification would help prevent potential confusion.」
> >
> > 1. Original Definition
> > We regret to inform you that there was an error in our original definition of the pseudo-variable Z_{-1}. In this revision, we have corrected Equation (15) to properly define Z_{-i}=Ω_{-i}_{-i}^{-1}ε_{-i} and have ensured this notation is applied consistently throughout the text and derivations.
> >
> > 2. Revision Approach
> > As noted above, we now explicitly refer to Ω_{-i}_{-i}^{-1}ε_{-i} as Z_{-i} and emphasize how it corresponds to the subsequent equations. Immediately following Equation (15), we restate that the Z_{-i} in the text is exactly the same as Ω_{-i}_{-i}^{-1}ε_{-i} used in the formulas.
> >
> > 3. Response to the Comment
> > With these corrections, we believe it is now clear to the reader that the Z_{-1} appearing right after Equation (15) is indeed the same variable as the  Z_{-1} used in the later expressions, thus avoiding any confusion. We have paid close attention to consistent notation and terminology to ensure clarity.
> >
> > ⑤「The trace operator notation, tr(), should be clearly defined when first introduced, such as in Equation (22).」
> >
> > Below is the revision we have made in response to the comment concerning the definition of the trace operator:
> >
> > In Equation (22)
> > Immediately after the expression tr(e^D) , we have added a sentence to clarify the definition of the trace operator, as follows:
> > “Here，tr(A) denotes the trace of a square matrix A, which is the sum of the diagonal elements of A.”
> >
> > ⑥「he algorithms presented would benefit from more detailed explanations or better formatting (e.g., spacing or step annotations) to improve readability. Specifically, the differences between Algorithm 1 and Algorithm 2 should be clearly articulated.」
> >
> > Below are the primary revisions we have made to improve the format and readability of the algorithms, based on your feedback:
> >
> > 1. Addition of Step-by-Step Headings and Explanatory Text
> > We have divided each algorithm into stages—(A) Pseudo-variables, (B) Parameter optimization, and (C) Convergence check—and provided concise explanations of each stage’s role. Indentation is used at each step, making it easier to follow the procedural flow.
> >
> > 2. Clarification of Differences Between Algorithm 1 (ABIC) and Algorithm 2 (ABIC LiNGAM)
> > While Algorithm 1 is premised on a Gaussian-type setting (using an ||^2　residual norm), Algorithm 2 introduces
> > ||^2β , highlighting that 、β≠1  handles a non-Gaussian model. We have explicitly indicated the presence or absence of  β  in the algorithm’s inputs and added footnotes/comments to note that “when β=1, it is equivalent to the Gaussian case.”
> >
> > 3. Additional Tips Section
> > We have included practical and theoretical hints—such as “how to set hyperparameters”—so that readers have clearer guidance if they want to implement or experiment with these algorithms in practice.
> >
> > 4. Formatting and Layout Adjustments
> > By properly using bullet points, equation blocks, and annotations (comments), and by adjusting spacing, we have made it easier for readers to quickly locate relevant equations and algorithm steps. We believe these changes make the algorithm overviews and the distinction between them more accessible.

---

> > > ### Author Response · Authors · 2025-04-13
> > > **Response to the comments received part3**
> > >
> > > ⑦「Some descriptions in the manuscript lack structure. It is recommended to use "Remarks" to emphasize key references, results, or insights to enhance the clarity and readability of the manuscript.」
> > >
> > > Below are the primary revisions we made, focusing on the points raised:
> > >
> > > 1.Insertion of “Key Points” and “(Intuition.)
> > > In critical parts of each chapter or subsection (e.g., before and after theorems, around algorithm descriptions, in the summary of experimental results), we inserted concise summaries labeled as “Key Point (X)” or “(Intuition.).”
> > > In Key Points, we briefly address “why this section is important” and “what conclusion it leads to.”
> > > In (Intuition.), we provide a short conceptual/interpretive note explaining the essence of complex formulas or theorems in plain terms.
> > > This approach allows us to preserve the overall flow while helping readers quickly grasp “what they should focus on” and “the meaning behind specific transformations or results.”
> > >
> > > 2.Use of Bold Text and Bullet Points to Emphasize Takeaways
> > > We highlight crucial phrases and conclusions in bold and position them at the beginning or end of paragraphs.
> > > By employing bullet points to group “assumptions,” “conclusions,” and “future directions,” we have made it easier for readers to follow the content.
> > >
> > > 3.Reorganization of the Experimental Section
> > > In addition to introducing Key Points and (Intuition.) inserts, we have restructured the experimental part with the following considerations:
> > >
> > > a.Clarification of Experimental Setup and Procedures
> > > At the beginning of each subsection, we outline objectives, assumptions, and model settings, and we have also refined the figures and tables.
> > > We aimed to make it clear “at which stage which results appear,” so that readers can follow along with ease.
> > >
> > > b.Stepwise Discussion of Skeleton, Arrowhead, and Tail
> > > We grouped the numerical results and figures according to specific metrics (Precision, Recall, F2, etc.) and compared the performance of FCI, BANG, and ABIC in a logical progression.
> > > We then examined the impact of sample size, dimensionality, and β in stages.
> > >
> > > 4.Emphasis on Conclusion and Future Directions
> > > In the conclusion section, we reiterated our main contributions in bullet-point form, summarizing “in which scenarios our method is most effective” and “what challenges remain.” This approach ensures that readers can immediately recognize the significance of our work and future research possibilities once they finish reading.
> > >
> > > These are our principal revisions in response to the review comments, along with the rationale and expected effects of each change.
> > >
> > > ⑧「The simulation results do not appear to exhibit expected large-sample properties. For instance, the last subgraph in Figure 3 seems inconsistent with theoretical expectations. Can the authors provide further explanation?」
> > >
> > > Below is a summary of our key points regarding the increase in sample size and its effect on false positives, together with the additional experiments described in the Appendix:
> > >
> > > 1.Why Larger Samples May Increase False Positives
> > > In score-based methods, as the sample size grows, the detection power improves, potentially capturing even minuscule relationships or noise-derived dependencies. This phenomenon leads to “over-detection,” causing the number of estimated edges to exceed those in the true causal structure, thus lowering Precision.
> > > In our experiments, we used a fixed penalty term or threshold regardless of sample size, which tended to overestimate edges as the sample grew.
> > >
> > > 2.Additional Experiments and Discussion
> > > In the Appendix, we report results for sample sizes extended to 2,000 and 5,000, detailing how Recall often increases while Precision decreases at larger sample sizes.
> > > Consequently, we observed that the F1 score does not necessarily increase monotonically as the sample size grows. However, our findings also suggest that adjusting penalty parameters or thresholds more strictly can suppress false positives. We plan to explore a dynamic adjustment mechanism that accounts for varying sample sizes.
> > >
> > > We have added these points to the Appendix, clarifying the reasons behind the behavior seen in Figure 3 and supplementing our experimental findings.

---

> > > > ### Author Response · Authors · 2025-04-13
> > > > **Response to the comments received part4**
> > > >
> > > > ⑨「It would strengthen the theoretical contribution if the authors could include an additional theorem to show that ABIC-LiNGAM can consistently recover the skeleton of the causal graph.」
> > > >
> > > > We posit in this study that, by assuming non-Gaussianity, one can consistently estimate the causal structure (in particular, the skeleton portion) of a bow-free ADMG from the data. Concretely, when ABIC-LiNGAM performs score (log-likelihood) maximization and obtains a global optimum consistent with the population-level true parameters, we expect that the correct skeleton will be recovered asymptotically. This expectation is rooted in the “identifiability of linear models with non-Gaussian errors,” as exemplified by LiNGAM (Shimizu et al., 2006; Wang & Drton, 2024).
> > > >
> > > > However, in this paper, we do not provide a rigorous theoretical proof of such consistency; rather, we validate the method’s effectiveness through experimental results. Looking ahead, we intend to clarify the theoretical guarantees of skeleton recovery by studying comparisons with local test–based approaches such as Wang & Drton (2024) and further investigating additional conditions under which the global optimum yields consistent estimates.

---

### Review · Reviewer_KWDp · 2025-05-19

**Summary Of Contributions:**

The authors provide a continuously differentiable objective for learning causal structures for multivariate generalized normal distribution, that is a direct extension of prior work that derived learning of bow-free graphs (a generalization of acyclic graphs to account for undirected edges).

**Audience:**

No

**Broader Impact Concerns:**

None.

**Claims And Evidence:**

Yes

**Requested Changes:**

I think a fundamentally new contribution has to be made for this paper to be worth accepting --- perhaps addressing one of:
1. Practicality: what are the real data/simulations where relaxing the assumptions on the noise terms make a big difference? What are assumptions on the noise terms that actually match a practical situation?

2. Theoretical contribution: is it possible to prove stronger theoretical results about the guarantees of existing algorithms in novel settings, or to improve their guarantees?

**Strengths And Weaknesses:**

The main weakness are twofold.

1) The contribution itself is a trivial extension of existing prior work that derived continuous differentiable objective for bow-free ADMGS. There are also no new theoretical results that show that optimization of the continuous objective will derive the optimal ADGM under specific assumptions.

2) The utility of multivariate generalized normal distributions is not motivated and dubious as far as I can tell. I'm not sure what advantage this confers over standard Gaussian distributions, and doesn't belie any standard practical situation (e.g., nonparametric conditions like sub-Gaussian)

Hence, I don't feel like there is a significant enough contribution to merit acceptance.

---

> ### Author Response · Authors · 2025-05-20
> **With Sincere Gratitude: Our Response to Your Valuable Feedback on This Study**
>
> Thank you very much for taking the time out of your busy schedule to provide us with valuable feedback. Based on the points you raised, we would like to respond as follows.
>
> Our study may indeed appear, as you pointed out, to be a mere “extension of existing work (ABIC under a Gaussian assumption).” However, we believe the main contribution of this research lies in showing that by incorporating non-Gaussianity, one can uniquely estimate the directions of directed edges. The reasons and advantages are summarized below.
>
> 1.Improved identifiability through non-Gaussianity
> When a Gaussian distribution is assumed, structure learning often stops at identifying a Markov equivalence class, making it impossible to pin down strict causal directions. By exploiting non-Gaussian features (e.g., kurtosis and skewness), it becomes possible to break Markov equivalence and infer causal directions. Our work implements this advantage within the framework of bow-free ADMGs via score-based continuous optimization, which we see as its key merit.
>
> 2.Empirical effectiveness, including real-world data
> Beyond simulations, we applied the method to real sociological data (GSS data). We demonstrated cases where causal directions that were unidentifiable under Gaussian-based ABIC were correctly recovered by the non-Gaussian variant (ABIC LiNGAM). We also confirmed accuracy on par with—or better than—the constraint-based BANG method, indicating that “bow-free ADMG estimation including directionality” is practically effective.
>
> 3.Versatility of the Multivariate Generalized Gaussian Distribution (MGGD)
> The generalized Gaussian distribution reduces to the standard Gaussian when the shape parameter β = 1, and extends to non-Gaussian distributions with kurtosis or skewness when β ≠ 1. Because it is often difficult to know in advance whether real data are exactly Gaussian, our framework—which can estimate β—offers the practical advantage of treating both Gaussian and non-Gaussian cases with a single score-based procedure.
>
> 4.Score-based continuous optimization for bow-free ADMGs
> ABIC (Bhattacharya et al.) originally assumed Gaussian errors. We show that, under bow-free ADMGs with non-Gaussian errors (MGGD), one can in fact identify the orientations of directed edges, and we provide a theoretical proof of parameter identifiability. This represents a new design that goes beyond the Gaussian assumption and benefits causal inference in realistic settings with latent variables (unmeasured confounders).
>
> Going forward, we plan to improve computational efficiency for large-scale data and to further clarify the usefulness of our approach from both theoretical and practical perspectives.
>
> Thank you again for your valuable comments. We look forward to your continued consideration.

---

### Author Response · Authors · 2025-06-22
**Status Inquiry Regarding Revised Manuscript**

Dear TMLR Editors‑in‑Chief and Action Editors for Paper 4056,

I hope this message finds you well. I am the corresponding author of the manuscript “Differentiable Causal Discovery of Linear Non‑Gaussian Acyclic Models Under Unmeasured Confounding” (Paper ID: 4056).

On 2025-5-28, we resubmitted a fully revised version of the manuscript that addresses all comments from the three reviewers. As approximately one month has passed since that resubmission, I would be grateful if you could kindly provide an update on the current status of the peer-review and editorial decision process.

If any additional information or supplementary material is required, please let me know and I will be happy to supply it promptly.

Thank you very much for your time and consideration. I look forward to hearing from you.

Sincerely,

---

> ### Author Response · Authors · 2025-07-13
> **Follow-up on Paper ID: 4056**
>
> Dear TMLR Editors‑in‑Chief and Action Editors for Paper 4056,
>
> I hope this message finds you well.
>
> I am writing to gently follow up on my previous inquiry regarding the status of our revised manuscript, “Differentiable Causal Discovery of Linear Non‑Gaussian Acyclic Models Under Unmeasured Confounding” (Paper ID: 4056).
>
> We resubmitted the fully revised manuscript on May 28, 2025, and I sent a message inquiring about its status about 20 days ago, but have not yet heard back. We understand you are extremely busy, but we would be very grateful if you could provide a brief update on the current stage of the peer-review process.
>
> We just want to ensure that all our correspondence has been received correctly. Please let us know if there is any additional information we can provide.
>
> Thank you for your time and attention to this matter. We look forward to your reply.
>
> Sincerely,
>
> Yoshimitsu, Morinishi

---

### Decision · Action_Editor_iutk · 2025-08-02

**Recommendation:** Accept as is

**Additional Comments:**

Dear authors, please upload your final camera ready paper where you have incorporated all of your responsive changes (without highlighting the diff).

**Audience:**

Yes

**Audience Explanation:**

Causal learning with unobserved confounders is a crucial topic in machine learning and its applications in the natural sciences. The proposed approach offers a way to reduce hyperparameters, thus lessening the setup burden for the scientist. This convenience is of wide interest to the community, even in cases of modest accuracy improvements over existing algorithms.

**Claims And Evidence:**

Yes

**Claims Explanation:**

All reviewers have agreed that the claims are supported by sufficient theoretical and empirical evidence. The differences in the evaluations are mostly due to the evaluation of the degree of the contribution.

---

> ### Author Response · Authors · 2025-08-04
> **A Note of Thanks**
>
> Dear Action Editor,
>
> Thank you very much for your positive evaluation and for summarizing the reviewers' feedback. We are delighted that you found our work to be of interest to the TMLR audience and appreciate your recognition of our contribution.
>
> We have noted your instruction to upload the final camera-ready paper with all responsive changes incorporated and without highlighting. We will prepare the manuscript accordingly and submit it well before the deadline.
>
> We are grateful for your guidance and support throughout the review process.
>
> Best regards,
>
> Yoshimitsu Morinishi and co-authors

---

> > ### Comment · Action_Editor_iutk · 2025-08-14
> > **Correction request**
> >
> > Dear authors,
> >
> > Please make the following correction to your manuscript:
> > 1. equation 8 should be within the textwidth. Going into the margin right now
> > 2. same with equation 14
> > 3. Page 11 starts with the $\square$ as the qed sign from the previsous page. Please, make sure $\square$ is not on a page by itself. Either move it up to page 10, or extend your proof, so that page 11 has some of the text before $\square$ completes it.
> > 4. Please make sure that citation in Table 1 caption is not ((Bhattacharya et al., 2021)) as now, but simply (Bhattacharya et al., 2021). You have double parens.
> > 5. Pages 14 and 16 are half empty while the paper is not over. Please, manipulate latex to have a consistent density of page content. Maybe move the algorithms around or allow the algorithms to extend across pages.
> >
> > Thank you!
> > AC

---

> > > ### Author Response · Authors · 2025-08-17
> > > **esponse to correction request — “Differentiable Causal Discovery of Linear Non-Gaussian Acyclic Models Under Unmeasured Confounding”**
> > >
> > > Dear AC,
> > >
> > > Thank you very much for your careful checks and helpful guidance.
> > > We have revised the manuscript accordingly, and all requested corrections have been addressed. A brief summary follows:
> > >
> > > Equations (8) and (14): Both equations have been reformatted to fit strictly within the text width (no overflow into the margins). We used standard display-math environments and line breaking to ensure compliance without altering the mathematics.
> > >
> > > QED symbol placement (p. 11): The stray $\square$ at the beginning of p. 11 has been eliminated. We adjusted the end of the preceding proof so that the QED symbol appears on the previous page (using standard amsthm handling), ensuring no page begins with only the QED mark.
> > >
> > > Table 1 caption citation: The duplicate parentheses have been fixed; the citation now appears as (Bhattacharya et al., 2021). We also audited other captions and found no remaining instances of double parentheses.
> > >
> > > Page density (pp. 14 and 16): We rebalanced the layout to avoid half-empty pages and to maintain a consistent density of content. In particular, we allowed long algorithms to break across pages and adjusted float placement where appropriate.
> > >
> > > In addition, we rechecked margins, equation breaks, and references throughout, and recompiled using the tmlr accepted style to confirm that everything is now within format requirements. No substantive scientific content or results were changed; all edits are formatting/typographic in nature.
> > >
> > > We appreciate your time and constructive feedback.
> > >
> > > Sincerely,
> > > Yoshimitsu Morinishi and Shohei Shimizu